# Improved Retrievals of Carbon Dioxide from the Orbiting Carbon Observatory-2 with the version 8 ACOS algorithm

Christopher W. O'Dell[1], Annmarie Eldering[2], Paul O. Wennberg[3], David Crisp[2], Michael R. Gunson[2], Brendan Fisher[2], Christian Frankenberg[3], Matthäus Kiel[3], Hannakaisa Lindqvist[4], Lukas Mandrake[2], Aronne Merrelli[5], Vijay Natraj[2], Robert R. Nelson[1], Gregory B. Osterman[2], Vivienne H. Payne[2], Thomas E. Taylor[1], Debra Wunch[6], Brian J. Drouin[2], Fabiano Oyafuso[2], Albert Chang[2], James McDuffie[2], Michael Smyth[2], David F. Baker[1], Sourish Basu[7,8], Frédéric Chevallier[9], Sean M. R. Crowell[10], Liang Feng[11,12], Paul I. Palmer[11,12], Mavendra Dubey[13], Omaira E. García[14], David W. T. Griffith[15], Frank Hase[16], Laura T. Iraci[17], Rigel Kivi[18], Isamu Morino[19], Justus Notholt[20], Hirofumi Ohyama[19], Christof Petri[20], Coleen M. Roehl[3], Mahesh K. Sha[21], Kimberly Strong[6], Ralf Sussmann[22], Yao Te[23], Osamu Uchino[19], and Voltaire A. Velazco[15]

[1]Colorado State University, Fort Collins, CO, USA
[2]Jet Propulsion Laboratory, California Institute of Technology, Pasadena, CA, USA
[3]California Institute of Technology, Pasadena, CA, USA
[4]Finnish Meteorological Institute, Helsinki, Finland
[5]SSEC, University of Wisconsin-Madison, Madison, WI, USA
[6]University of Toronto, Toronto, Canada
[7]NOAA Earth System Research Laboratory, Global Monitoring Division, Boulder, CO, USA
[8]Cooperative Institute for Research in Environmental Sciences, University of Colorado BoulderBoulder, Colorado, USA
[9]Laboratoire des Sciences du Climat et de l'Environnement, IPSL, CEA-CNRS-UVSQ, Gif-sur-Yvette, France
[10]College of Atmospheric and Geographic Sciences, University of Oklahoma, Norman, OK, USA
[11]National Centre for Earth Observation, University of Edinburgh, UK
[12]School of GeoSciences, University of Edinburgh, UK
[13]Los Alamos National Laboratory, Los Alamos, NM, USA
[14]Izaña Atmospheric Research Center, Meteorological State Agency of Spain (AEMet), Santa Cruz de Tenerife, Spain
[15]Centre for Atmospheric Chemistry, University of Wollongong, Wollongong, Australia
[16]Karlsruhe Institute of Technology, IMK-ASF, Karlsruhe, Germany
[17]NASA Ames Research Center, Moffett Field, CA, USA
[18]Finnish Meteorological Institute, Sodankylä, Finland
[19]National Institute for Environmental Studies (NIES), Tsukuba, Japan
[20]University of Bremen, Bremen, Germany
[21]Royal Belgian Institute for Space Aeronomy, Brussels, Belgium
[22]Karlsruhe Institute of Technology, IMK-IFU, Garmisch-Partenkirchen, Germany
[23]LERMA-IPSL, Sorbonne Université, Observatoire de Paris, Université PSL, CNRS, F-75005, Paris, France

*Correspondence to:* Christopher W. O'Dell
(Christopher.ODell@colostate.edu)

**Abstract.** Since September 2014, NASA's Orbiting Carbon Observatory-2 (OCO-2) satellite has been taking measurements of reflected solar spectra and using them to infer atmospheric carbon dioxide levels. This work provides details of the OCO-2 retrieval algorithm, versions 7 and 8, used to derive the column-averaged dry air mole fraction of atmospheric $CO_2$ ($X_{CO_2}$) for the roughly 100,000 cloud-free measurements recorded by OCO-2 each day. The algorithm is based on the Atmospheric

Carbon Observations from Space (ACOS) algorithm which has been applied to observations from the Greenhouse Gases Observing SATellite (GOSAT) since 2009, with modifications necessary for OCO-2. Because high accuracy, better than 0.25%, is required in order to accurately infer carbon sources and sinks from $X_{CO_2}$, significant errors and regional-scale biases in the measurements must be minimized. We discuss efforts to filter out poor quality measurements, and correct the remaining good-quality measurements to minimize regional-scale biases. Updates to the radiance calibration and retrieval forward model in version 8 have improved many aspects of the retrieved data products. The version 8 data appear to have reduced regional-scale biases overall, and demonstrate a clear improvement over the version 7 data. In particular, error variance with respect to TCCON was reduced by 20% over land and 40% over ocean between versions 7 and 8, and nadir and glint observations over land are now more consistent. While this paper documents the significant improvements in the ACOS algorithm, it will continue to evolve and improve as the $CO_2$ data record continues to expand.

## 1 Introduction

Bias-free measurement of atmospheric $CO_2$ concentrations from space is a long-pursued goal in the carbon cycle community. Such measurements are critical for inferring sources and sinks of carbon, and how these sources and sinks change over time due to both anthropogenic and natural causes (e.g. Rayner and O'Brien, 2001; Chevallier et al., 2007; Baker et al., 2010). The first instrument capable of $CO_2$ measurements from space using the near- and short-wavelength infrared was SCIAMACHY, the SCanning Imaging Absorption spectroMeter for Atmospheric CHartographY (Buchwitz et al., 2005; Reuter et al., 2011), which operated from 2002 to 2012. This was followed by the first dedicated greenhouse gas satellite, the Japanese Greenhouse gases Observing SATellite (GOSAT), which launched in January 2009 (Yokota et al., 2009). The Orbiting Carbon Observatory-2 (OCO-2) followed on July 2, 2014, with the goal of measuring the column-averaged dry air mole fraction of carbon dioxide ($X_{CO_2}$) with sufficient precision and accuracy to enable greatly enhanced understanding of the surface-atmosphere exchange of $CO_2$ on regional scales (Crisp et al., 2008; Crisp, 2015). OCO-2 was preceded by the original OCO mission, which failed due to a launch vehicle malfunction in 2009. Retrieval algorithms originally developed for OCO (Connor et al., 2008) have been continuously refined since 2009 (O'Dell et al., 2012), by application to data from GOSAT.

$X_{CO_2}$ measurements from the OCO-2 version 7 data product (Eldering et al., 2017) have recently been used to estimate $CO_2$ fluxes from both natural (Liu et al., 2017; Chatterjee et al., 2017; Crowell et al., 2018a) and anthropogenic (Hakkarainen et al., 2016; Schwandner et al., 2017; Nassar et al., 2017) sources; see Eldering et al. (2017) for a complete review of these findings. However, $X_{CO_2}$ measurements must be both extremely accurate and precise in order to accurately determine fluxes (Miller et al., 2007), since fluxes are determined from small (<2.5%) spatial and temporal gradients in the $X_{CO_2}$ field. Spatially coherent biases in $X_{CO_2}$ on regional scales as small as a few tenths of a part-per-million (ppm) in $X_{CO_2}$ can lead to spurious values of inferred fluxes (Chevallier et al., 2014).

**Table 1.** Prescreening filter criteria.

| Category | Land Criterion | Ocean Criterion |
|---|---|---|
| Successful Measurement | Sounding_Qual_Flag $= 0$ | Same as land |
| A-Band Preprocessor | Cloud_Flag $= 0$ | Same as land |
| Solar Geometry | SZA $< 85°$(nadir), $< 80°$(glint) | Same as land |
| IMAP Preprocessor | $0.985 <$ co2_ratio $< 1.045$ | Same as land |
| Band 1 SNR | $SNR_1 \geq 100$ | Same as land |
| Band 3 SNR | $SNR_3 \geq 75$ | Same as land |
| Land Fraction | $f_{land} \geq 80\%$ | $f_{land} \leq 20\%$ |

The ACOS algorithm was originally developed for OCO. It was first applied to GOSAT data in 2009 and has continuously evolved and improved in the intervening years. Generally, good error statistics were shown for GOSAT observations over both land and water, with typical biases below 1 ppm based on comparisons to both ground-based (Lindqvist et al., 2015; Kulawik et al., 2016) and aircraft (Frankenberg et al., 2016) validation data. After the successful launch of OCO-2, the ACOS algorithm
was further modified and tuned for application to the OCO-2 spectra. $X_{CO_2}$ error statistics are similar to those from GOSAT, with RMS errors less than 1.5 ppm when compared against most ground-based Total Carbon Column Observing Network (TC-CON, Wunch et al. (2010)) stations (Wunch et al., 2017). However, Wunch et al. (2017) noted that important biases remain, in particular related to latitude, surface properties, and atmospheric scattering by clouds and aerosols. A particularly troubling bias evident in the southern hemisphere mid-latitude ocean in austral winter had amplitudes as large as several ppm. This bias
was not seen in ACOS retrievals using GOSAT data, though GOSAT's ocean glint viewing geometry was restricted and could not typically see this far south, potentially masking the problem.

The primary purpose of this paper is to describe the details of the ACOS $X_{CO_2}$ retrieval algorithm as applied to OCO-2 data, in particular the latest version 8 (also referred to as build 8 or B8). Because science results have already been published
with version 7 (also referred to as build 7 or B7) as discussed above, we also discuss the differences between versions 7 and 8. This paper is organized as follows: Section 2 discusses prescreening of the data to remove cloudy and difficult-to-retrieve soundings. Section 3 lists the details of the retrieval algorithm and its evolution since O'Dell et al. (2012). Section 4 discusses the methodology and results of the post-retrieval filtering and bias correction. Section 5 provides a brief evaluation of $X_{CO_2}$ from both versions 7 and 8, and the discussion in Section 6 concludes the paper.

## 2 Data and Prescreening

Because only scenes with sufficient signal and nearly devoid of cloud and aerosol contamination can yield successful $X_{CO_2}$ retrievals, a prescreener is used for OCO-2 soundings before processing by the Level-2 "Full-Physics" (L2FP) $X_{CO_2}$ retrieval algorithm. Our prescreening module requires outputs from two fast algorithms, described in detail in Taylor et al. (2016). First, the "A-band Preprocessor" (ABP) performs a fast retrieval of surface pressure using the $O_2A$ band only, assuming that no clouds or aerosols are present. Poor spectral fits and differences between the retrieved and *a priori* surface pressure greater than 25 hPa are used to identify the presence of cloud or aerosol contamination. Scenes without sufficient signal-to-noise in the $O_2A$ band are skipped altogether. Second, the "IMAP-DOAS" preprocessor performs fast, clear-sky fits to the weak and strong $CO_2$ bands at 1.61 and 2.06 $\mu$m, respectively. While this preprocessor solves for a number of variables, the $CO_2$ and $H_2O$ columns, which are fit independently from each of these two bands, are most relevant for cloud screening. From these spectral fits, the strong-to-weak ratios of the column-integrated $CO_2$ and $H_2O$ are derived. The $CO_2$ ratio must be within a certain range (near unity) for the scene to be deemed sufficiently clear to warrant a Full-Physics retrieval. Other screens are used to remove soundings unlikely to yield successful $X_{CO_2}$, such as those at high solar zenith angle or for which the continuum SNR levels are too low. Unlike in version 7 of the OCO-2 algorithm, there is no explicit screen for snow and ice-covered surfaces. However, the surface albedo in the strong $CO_2$ band is low over snow and ice, and therefore the strong $CO_2$ band SNR filter will remove many of those scenes. The full prescreening criteria for OCO-2 B8 are given in Table 1.

In total, roughly 26% of land soundings pass our pre-screener (28% land nadir, 25% land glint) and 27% of ocean glint soundings pass it as well. Generally these fractions are strong functions of both location and time of year. To illustrate this, the fraction of soundings passing the prescreening criteria for December 2015 and June 2016 are shown in Figure 1. A number of features are observed. A higher fraction of soundings are passed in the tropics than at higher latitudes relative to the sub-solar latitude ( ∼-23° in December and +23° in June), and the passing rates tend to be higher over bright vs. dark surfaces. Also, few soundings survive over the tropical rainforests in South America and Africa, which are often cloudy. A significant number of soundings survive prescreening over the Greenland and Antarctic ice sheets during their summer season (this was not the case in version 7), though it is shown later that most of these fail the post-retrieval quality screening (Section 4.2). About 10% of nadir soundings over ocean pass the prescreening criteria; this occurs in regions where the nadir view is relatively close to the glint geometry, typically near the sub-solar latitude. These nadir ocean soundings are currently removed by post-retrieval filtering, as their quality relative to the glint ocean observations has not yet been evaluated. A final obvious feature is that fewer soundings are available in nadir mode than in glint - this is because many orbits over the Atlantic and Pacific oceans became "full-time" glint-mode orbits beginning in November 2015 (Crisp et al., 2017). Prior to that, there were equal numbers of nadir and glint orbits, but after that change, approximately one third of all orbits are nadir and two thirds are glint.

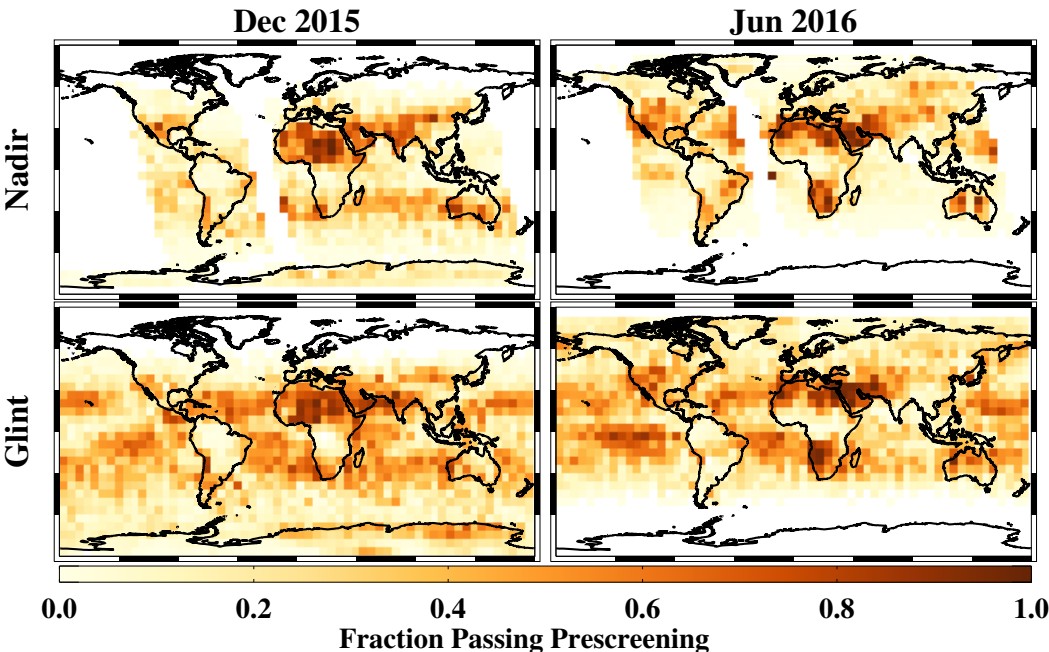

**Figure 1.** Fraction of Soundings passing OCO-2 B8 prescreening filter in December 2015 (left) and June 2016 (right), for both nadir mode (top) and glint mode (bottom). Starting in November 2015, about one third of all orbits are performed in nadir mode, and two thirds are performed in glint mode.

## 3   The NASA ACOS $X_{CO_2}$ retrieval algorithm as applied to OCO-2

The original ACOS $X_{CO_2}$ retrieval algorithm over land (version 2.9) was described in O'Dell et al. (2012), with details specific to GOSAT given in Crisp et al. (2012). Details of the spectroscopy used at that time were published in Thompson et al. (2012). In this section, we give an overview of the evolution from ACOS version 2.9 to OCO-2 versions 7 and 8, including

5    spectroscopy, aerosol treatment, and a number of other changes.

Briefly, the NASA ACOS algorithm uses optimal estimation to solve for parameters of a state vector to obtain the best match to spectra from the three GOSAT or OCO-2 near- infrared bands and consistent with a prior constraint. These bands are the $O_2A$ band at 0.76 $\mu$m (band 1), the weak $CO_2$ band at 1.61 $\mu$m (band 2), and the strong $CO_2$ band at 2.06 $\mu$m (band 3). The

10   state vector parameters, listed in Table 2, include the profile of $CO_2$ at twenty atmospheric levels along with a number of ancillary parameters to which the GOSAT and OCO-2 near-infrared spectra are sensitive. These include surface pressure, surface albedo parameters (over land only), a temperature profile offset and water vapor profile multiplier, and parameters related to the wavelength scale of the spectra (dispersion shift and stretch). The latter are relative to the preflight values of these parameters, described in Lee et al. (2017). Because telluric line positions are known with high accuracy, the retrieval solves for them with

**Table 2.** General setup of the ACOS state vector.

| Element | No. Elements | Prior Value | Prior Uncertainty ($1\sigma$) | Notes |
|---|---|---|---|---|
| $CO_2$ Values | 20 | Same as TCCON | Same as ACOS B2.9 | Defined on sigma pressure levels |
| Temperature Offset | 1 | 0 K | 5 K | Rel. to Prior profile |
| Surface Pressure | 1 | from prior meteorology | 4 hPa | Prior Unc. 1 hPa for B3.5 |
| $H_2O$ Scale Factor | 1 | 1.0 | 0.5 | Multiplier on prior profile |
| Aerosol Type 1,2 $OD_{755}$ | 2 | from MERRA | $\pm$ factor of 7.39 | |
| Water, Ice Cloud $OD_{755}$ | 2 | 0.0125 | $\pm$ factor of 6.05 | |
| Aerosol Type 1,2 $x_0$ | 2 | 0.9 | 0.2 | |
| Water Cloud $x_0$ | 1 | 0.75 | 0.4 | |
| Ice Cloud $x_0$ | 1 | just below tropopause | 0.2 | |
| Aerosol Type 1,2 $\sigma_a$ | 2 | 0.05 | 0.01 | |
| Water Cloud $\sigma_a$ | 1 | 0.1 | 0.01 | |
| Ice Cloud $\sigma_a$ | 1 | 0.04 | 0.01 | |
| UTLS Aerosol $OD_{755}$ | 1 | 0.006 | $\pm$ factor of 6.05 | introduced in B8 |
| Albedo Mean Land | 1 per band | Prior Calc. | 1.0 | Land |
| Albedo Slope Land | 1 per band | 0.0 | 0.0005 | Land; units of $1/cm^{-1}$ |
| Albedo Mean Ocean | 1 per band | 0.02 | {0.2,0.2,1e-3} | Ocean |
| Albedo Slope Ocean | 1 per band | 0.0 | 1.0 | Ocean; units of $1/cm^{-1}$ |
| SIF Mean | 1 | Prior Calc. | 0.008 | Land |
| SIF Slope | 1 | 0.0018 | 0.0007 | Land; units of $1/cm^{-1}$ |
| Wind Speed | 1 | from prior meteorology | 5 m/s | Ocean |
| Dispersion Shift | 1 per band | 0.0 | 0.4 of channel FWHM | |
| Dispersion Stretch | 1 per band | 0.0 | 1 pm/channel | OCO-2 only |
| EOF Amplitudes | 3 per band | 0.0 | 10.0 | 1 per band for B3.5 & earlier |

virtually no dependence on the prior. To account for scattering effects of thin cloud or aerosol, the retrieval also solves simultaneously for amounts and Gaussian vertical profiles (as described in Section 3.1) of five different kinds of scatterers with fixed optical properties: a water cloud type, an ice cloud type, two fixed aerosol types, and beginning in version 8, an Upper Tropospheric/Lower Stratospheric (UTLS) sulfate aerosol layer. In addition, the retrieval also fits scaling factors for three spectral patterns per band, to account for imperfections in the spectroscopy, solar model, and instrument model, and determined using singular value decomposition of our fit residuals run on clear-sky soundings (Section 3.3). For solar-induced fluorescence (SIF) emission from plants on land, we fit for two SIF parameters which are needed to account for this fluorescence in the L2 spectra (Section 3.5). These SIF parameters are not the official SIF data product; that product is derived from the IMAP prescreener

**Table 3.** Significant ACOS retrieval algorithm changes.

| GOSAT B2.10 | GOSAT B3.3 | GOSAT B3.4 | GOSAT B3.5 |
|---|---|---|---|
| Gaussian Aerosol Profiles | Residual Fitting | Updated ocean surface | MERRA Aerosol Types |
| Sigma Pressure Levels | 1 hPa $P_{surf}$ Prior Uncertainty | Band 2 Spectral Range | |
| Prior $CO_2$ Profile Change | Prior $OD_{755} = 0.05$ | Spectroscopy Update | |
| Spectroscopy Update | Spectroscopy Update | | |
| Corrected $X_{CO_2}$ AK | Fluorescence Fit Land Gain H | | |

| GOSAT B7.3 | OCO-2 B7 | OCO-2 B8 | |
|---|---|---|---|
| 3 EOFs per band | Restricted Band Ranges | Spectroscopy Update | BRDF over land |
| 2 hPa $P_{surf}$ Prior Uncertainty | 4 hPa $P_{surf}$ Prior Uncertainty | UTLS Aerosol | GEOS5-FP-IT Meteorology |
| Updated cloud ice properties | | L1B improvements | numerous small changes |

**Table 4.** ACOS retrieval differences between GOSAT and OCO-2.

| Category | GOSAT | OCO-2 |
|---|---|---|
| Radiance used | Estimated total intensity | OCO-2 single polarization |
| EOFs, Band ranges | wavenumber space | channel space |
| Fit $O_2A$ band offset ? | Yes | No |
| SIF Prior | 0 | From IMAP retrieval |
| Per-band dispersion parameters | Offset only | Offset, Slope |
| Band 1 Fitted Range | 758.1–772.2 nm | 759.2–771.5 nm |
| Band 2 Fitted Range | 1597.4–1618.1 nm | 1598.1–1617.9 nm |
| Band 3 Fitted Range | 2042.1–2079.0 nm | 2047.8–2079.9 nm |
| Channel Mask | None | Bad samples, spikes |

through a dedicated fit (Sun et al., 2018). In total, there are typically 55 fitted parameters for land retrievals and 53 for ocean[1].
With the exception of $CO_2$, the *a priori* covariance matrix is diagonal, with the $1\sigma$ uncertainties as given in Table 2.

The first documented algorithm version, B2.9 as described in O'Dell et al. (2012), had several deficiencies which occasion-
5   ally produced large biases in the retrieved $X_{CO_2}$ (Wunch et al., 2011a). This early version of the algorithm also contained some

---
[1]This excludes parameters in our state vector with prior uncertainties close to zero, such as cloud and aerosol layer widths.

cumbersome traits, such as a variable number of vertical levels from sounding to sounding, which made the output difficult to use. The observed $X_{CO_2}$ biases were partially related to the aerosol parameterization, demonstrated by the fact that clear-sky retrievals of clear-sky simulations did not exhibit substantial biases (O'Dell et al., 2012). Furthermore, errors in the $O_2$ and $CO_2$ spectroscopy were suspected to be an additional source of bias. Over the course of several years, a number of changes to the algorithm were therefore implemented to yield the present version B8. The changes are too numerous to fully describe here, but the most important ones are listed in Table 3. The changes fall into several major categories, with spectroscopy, aerosol treatment, treatment of the ocean surface, and chlorophyll fluorescence being the most important. In B8, the meteorology used to prescribe the a priori temperature profile, water vapor profile, and surface pressure was also changed (Section 3.5).

Further, as listed in Table 4, some minor retrieval differences exist between the GOSAT and OCO-2 versions of the algorithm. Besides using instrument models specific to each instrument (such as wavelengths of the various channels, noise model, and instrument line shape functions), slightly different spectral ranges are fit for each instrument. Generally, this is because the trusted calibrated range of OCO-2 spectra is slightly smaller than that of GOSAT, due to the differences in design of the OCO-2 grating spectrometer versus the GOSAT Fourier transform spectrometers. Additionally, while all channels in each band in the given spectral ranges are used for GOSAT, some band channels are masked out for OCO-2. This is due to either underlying bad pixels in the detector arrays, or to transient cosmic rays that induce temporary spurious readings in random channels. Both of these processes are described in detail in Crisp et al. (2017).

## 3.1 Aerosol-related changes

Starting with version B2.10, the 20-layer optical depth retrieval used for clouds and aerosols was replaced with a Gaussian-shaped vertical profile for each of the retrieved scattering particle types. As of version 8, two cloud types, two lower-atmosphere aerosol types, and one stratospheric aerosol are used. The new cloud and aerosol profile treatment is similar to that of Butz et al. (2009) but specifies the aerosol concentration $\rho_{aer}$ as a function of $x$, the pressure relative to the surface pressure. Therefore, $x$ ranges from zero at the top of the atmosphere to one at the surface. The functional form is simply

$$\rho_{aer}(x) = C \exp\left(-\frac{(x-x_0)^2}{2\sigma_a^2}\right) \tag{1}$$

where for each aerosol type $x_0$ is the vertical location at peak aerosol density and $\sigma_a$ is the Gaussian $1\sigma$ profile width. Both of the latter variables are specified in units of relative pressure $x$. The prefactor $C$ is defined such that the aerosol or cloud optical depth at 755 nm, hereafter $OD_{755}$, equals the desired value. In the retrieval algorithm, the fitted quantities are $\ln OD_{755}$ and peak height $p_{r,0}$ for each aerosol type, with the exception of the stratospheric aerosol (described in Section 3.1.1) for which only the optical depth is retrieved. Because it has been shown that GOSAT and OCO-2 -like spectra have little sensitivity to the Gaussian profile width (Butz et al., 2009), this parameter is fixed in both the GOSAT and OCO-2 retrievals for all particle types. The prior profiles for each fitted type are shown in Figure 2.

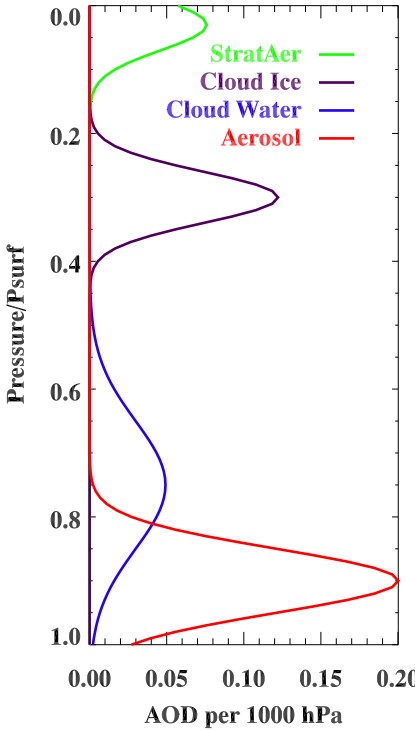

**Figure 2.** Prior Gaussian profiles of the lower tropospheric aerosol types (red), water cloud (blue), ice cloud (purple), and stratospheric aerosol (green). The local aerosol optical depth (AOD) per unit pressure at 755 nm is plotted as a function of the relative pressure. The lower tropospheric aerosol prior optical depth is not fixed as for the other types, but rather is taken from a climatology described in the text.

The change to a sigma-level pressure system was incorporated at about the same time as the shift to Gaussian aerosol profiles. Instead of fixed pressure levels, the pressure levels scale with the surface pressure:

$$p_i = a_i \, p_{surf} \tag{2}$$

where the $a_i$ are chosen such that the total number of pressure boundaries is 20, and the layers have roughly equal pressure widths. The top-most model level is set to 0.01 hPa.

The optical properties of the four scattering types remained unchanged from version B2.9 to B3.4 and are described in O'Dell et al. (2012). However, the use of two fixed aerosol types, type "2b" and "3b" from the Kahn et al. (2001) climatology, did not accurately represent the true global variability of aerosol on the length and time scales probed by GOSAT and OCO-2. Beginning with build 3.5, the aerosol types were changed to be location and time dependent, with the prior type information coming from the aerosol climatology of the Modern-Era Retrospective analysis for Research and Applications (MERRA,

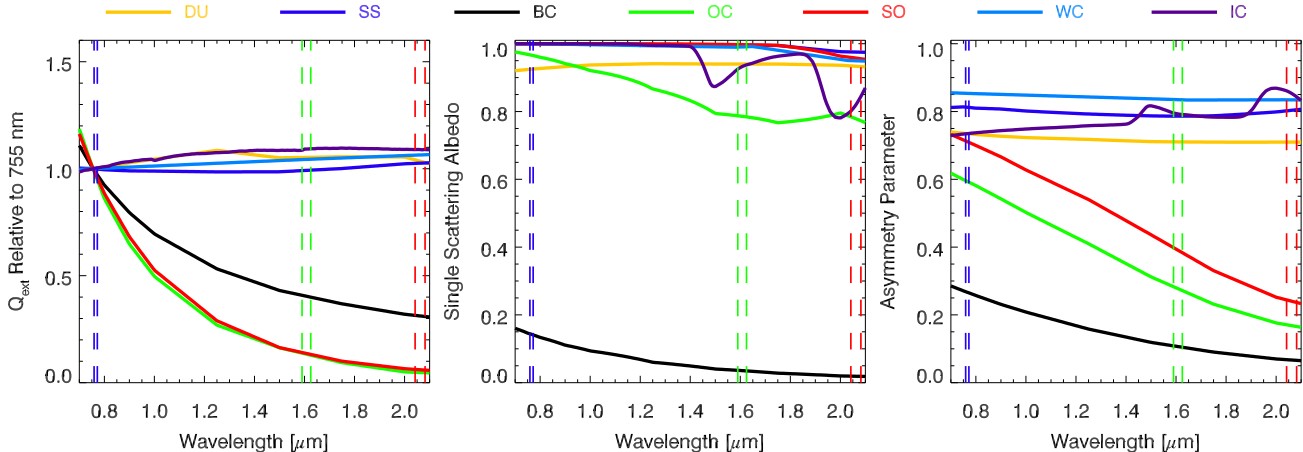

**Figure 3.** Optical properties of aerosols and clouds used in the L2FP code as a function of wavelength. Left: Extinction efficiency relative to that at 755 nm. Middle: Single scattering albedo. Right: Asymmetry parameter. DU: Dust, SS: Sea Salt, BC: Black Carbon, OC: Organic Carbon, SO: Sulfate, WC: Water Cloud, IC: Ice Cloud. The spectral ranges of the three OCO-2 bands are demarcated by the dashed vertical lines.

Rienecker et al. (2011)). The MERRA aerosol field is driven by the Georgia Tech/Goddard Global Ozone Chemistry Aerosol Radiation and Transport (GOCART) model (Chin et al., 2002), and modified by assimilating aerosol optical depth from the MODIS instruments onboard the *Terra* and *Aqua* satellites (Colarco et al., 2010). MERRA contains five broad aerosol types: dust (DU), sea salt (SS), sulfate (SO), and black and organic carbon (BC and OC, respectively). Dust and sea salt are each

tracked in five separate size bins. Organic and black carbon are tracked in both hydrophobic and hydrophilic categories. In addition to the carbonaceous types, sulfate aerosol and sea salt are also hydrophilic and hence have optical properties that depend on the local relative humidity (RH).

For the aerosol prior in the ACOS retrieval, we primarily sought to specify the typical dominant aerosol types present (in

terms of their contribution to the optical depth in the OCO-2 bands) in a given location at a given time of year. Monthly aerosol fields were derived from the MERRA model for the year 2010, and are used for all years in the ACOS retrieval. We aggregated the 15 MERRA types, eight of which have RH-dependent optical properties, into the five aggregated types listed above. We used typical density weightings and relative humidity values to create the optical properties for these aggregated types, as described in Crisp et al. (2010). Their optical properties, including extinction efficiency, single scattering albedo, and asymmetry

parameter, are shown in Figure 3. The organic carbon and sulfate aerosol are generally similar in their optical properties, though their single scattering albedos diverge somewhat in the $CO_2$ bands. The sea salt, water cloud, and dust optical properties are relatively similar across the OCO-2 spectral range.

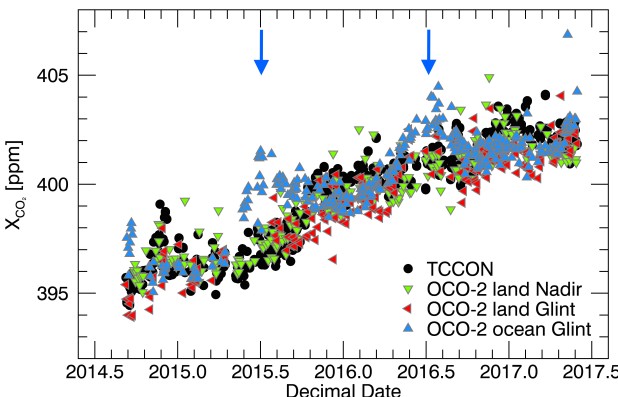

**Figure 4.** Comparison of $X_{CO_2}$ time series for OCO-2 version 7 and TCCON, over several years at the station in Wollongong, Australia (Griffith et al., 2014b). Each OCO-2 symbol represents an overpass average. A simple geometric colocation strategy was used in which OCO-2 soundings within $\pm 7.5°$ latitude and $\pm 30°$ longitude of the TCCON station were retained. Large positive biases occur in the ocean glint soundings in the southern hemisphere winter (blue arrows). As seen in Figure 19, these large biases primarily occur in the southern oceans.

At each sounding location, the two aggregated aerosol types with the highest mean monthly values of the $OD_{755}$ are selected to be retrieved by the L2FP algorithm. In previous algorithm versions, the total prior $OD_{755}$ was set to 0.15, apportioned equally among four scattering types (water cloud, ice cloud, and two tropospheric aerosol types). However, it was found this was generally too high to allow a fit near $OD_{755}$=0 for scenes that were almost entirely free of aerosol. This "clear-sky bias"

was seen in early simulation tests (O'Dell et al., 2012). The prior $OD_{755}$ is now set to 0.0125 for each cloud type, and set from the MERRA aerosol climatology for each tropospheric aerosol type as the average $OD_{755}$ of that type (at a particular location and month). There is some evidence that the tropospheric aerosol priors are occasionally still too high; methods for specifying the aerosol prior are a continuing topic of investigation.

The cloud ice optical properties were updated in version 7. Before that, they were based on the band-averaged model developed by Baum et al. (2005) primarily for the MODIS instrument and known as the MODIS Collection 5 model. This cloud ice model considered an ensemble of size-dependent non-spherical ice crystal habits in random orientation. As ice crystal surface roughness was later shown to significantly affect scattering by ice crystals, and simulations with roughened model particles were more consistent with satellite observations of ice cloud polarized reflectances (Yang et al., 2013), we

updated the cloud ice optical properties to correspond to the MODIS Collection 6 model, which describes scattering by severely roughened hexagonal column ice crystal aggregates (Baum et al., 2014). This update also fixed several minor issues in the

previous cloud ice model, such as those resulting from linear interpolation of the optical properties from MODIS wavelength bands to OCO-2, and those relating to truncation of the phase function.

### 3.1.1 The need for a stratospheric aerosol

When validating version 7 $X_{CO_2}$ retrievals, it was discovered via comparisons to both TCCON and models that most ocean soundings in the most southerly ~10 degrees of latitude exhibited a high bias of 1-3 ppm during the austral winter (Wunch et al., 2017). Figure 4 shows the bias appear in the southern hemisphere winter over the Wollongong TCCON station. The bias is seen in soundings over ocean but not land. The bias is also apparent relative to the Lauder TCCON station (Figures 11 and A1 from Wunch et al., 2017). Comparisons of OCO-2 soundings to models (Figure 19) show the bias as a quasi-zonal band over the southern hemisphere oceans, again with the larger bias occurring in the southern hemisphere winter. There is also evidence of a similar but weaker band of high bias in the northern hemisphere. For 2015, it was hypothesized that small aerosol particles may have been injected into the UTLS by the explosive eruptions of the Calbuco (22-30 April 2015) and Wolf (late May 2015) volcanos in south-central Chile and the Galapagos Islands, respectively. The presence of an aerosol layer with visible optical depths around 0.01 was later confirmed with observations from the Cloud-Aerosol Lidar and Infrared Pathfinder Satellite Observatory (CALIPSO) and the Ozone Mapping Profile Suite (OMPS) satellites (Bègue et al., 2017). These optical depths are small, but have a large impact on the radiances, especially in the $O_2A$ band, due to their high altitude.

It was recognized that our version 7 retrieval algorithm had no way to accommodate the spectral signature of small stratospheric aerosol particles, which have a significantly larger effect on the $O_2A$ band than either of the $CO_2$ bands due to the small size parameter, i.e, the ratio of the size of the scattering particle to the spectral wavelength. The spectral signature would essentially appear as a radiance offset in the $O_2A$ band. As a first test, we ran hundreds of retrievals on a single sounding that had a large positive bias in the operational retrieval, using slightly different first-guess values for each retrieval. Essentially, a continuum of solutions was found (Figure 5). On one end of retrieval space, an approximately correct value of surface pressure was found by inserting a thicker ice cloud, which contains larger particles starting in the stratosphere, and therefore has a strong effect on all three bands (see Figure 3). This type of solution produced a poor $\chi^2$ in the strong $CO_2$ band, typically $> 2$. On the other end of the continuum were solutions where the sulfate layer, which was placed near the surface in the prior, was moved high up into the atmosphere. This solution regime had a much lower reduced $\chi^2$ (around 1.5) in the strong $CO_2$ band and an $X_{CO_2}$ that was typically 3–4 ppm lower, and much more in line with TCCON and model estimates. In these cases, the amount of upper atmosphere cloud ice retrieved was also reduced, as its role was taken over by the sulfate.

These tests indicated that a more realistic solution would often be found if the retrieval could push the prior sulfate into the upper atmosphere, though this seldom occurred. The amount of sulfate needed in the upper atmosphere in these cases is small, approximately 0.01 optical depth at 755 nm. That value is consistent with other observations (Bègue et al., 2017). In addition to actual small particles in the UTLS, the OCO-2 instrument has a documented problem which produces a similar impact on the $O_2A$ and spectrum. As described in Crisp et al. (2017, Section 6.5), a very thin layer of ice appears to build up on the OCO-2

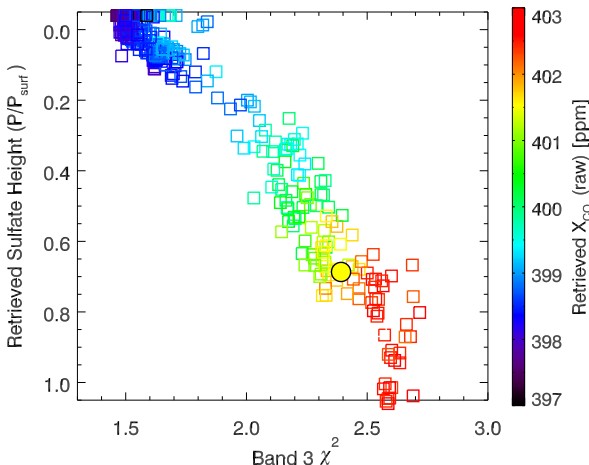

**Figure 5.** Results of several hundred retrievals of a single ocean glint sounding (28.5°S, 52.3°W) measured on June 26, 2015. Each retrieval is identical except that each has a different first guess state, consistent with the prior uncertainty distribution. The retrieved relative sulfate height (0=top-of-atmosphere; 1=surface) is shown on the ordinate, the reduced $\chi^2$ from the strong (2.06 $\mu$m) $CO_2$ band retrieval is shown on the abscissa, and the retrieved $X_{CO_2}$ is indicated by color. For reference, the result from the operational retrieval (version 7) is shown as the large filled circle. When the retrieval places the sulfate near the surface, as in the version 7 case, both the strong $CO_2$ band $\chi^2$ value and $X_{CO_2}$ tend to be higher. Conversely, when the retrieval pushes the sulfate closer to the top-of-atmosphere, the strong $CO_2$ band $\chi^2$ values and $X_{CO_2}$ tend toward lower values, a result that is more physically plausible.

Focal Plane Arrays (FPAs) over time. As this ice layer grows to a thickness similar to the anti-reflective coating thickness (tens of nanometers), it enhances the surface reflectance on the $O_2A$ band FPA, producing a scattered light contribution of 0.1 to 0.2%. Much smaller effects are seen on the $CO_2$ detectors. The ice layer is sublimed off every 3-6 months when the instrument goes through a decontamination cycle. While attempts have been made to remove this scattered light contribution in the version

5    8 calibrated radiance (L1B) product, it is likely that some residual signal remains. Because this is primarily a radiance offset in the $O_2A$ band alone, it produces a signal similar to a small UTLS aerosol, and hence would also be mitigated by including a stratospheric aerosol in the retrieval. During algorithm testing of the stratospheric aerosol using version 7 L1B radiances (which contained the scattered light signature), we found that the amount of UTLS aerosol retrieved indeed correlated with the decontamination cycles, lending credence to this hypothesis.

    Thus, in version 8 an additional sulfate aerosol was included in the retrieval state vector. For simplicity, a sulfate type identical to the lower-atmosphere type in terms of optical properties was used. Only the total optical depth of the stratospheric sulfate is retrieved, while its Gaussian height and width are kept fixed. This solutions treats both actual small particles in the UTLS as well as the radiometric offsets that accompany the real $O_2A$ band scattered light signal. Our testing of the version

8 algorithm showed that including this state vector element not only reduced the southern ocean bias, but also reduced the negative tropical ocean bias and positive bias over higher northern latitude lands that were also apparent in Figure 19. A more complete comparison of version 7 and version 8 validation statistics is given in Section 5.

### 3.2 Spectroscopy-related changes

There have been substantial changes between the molecular cross sections used in the earliest ACOS versions and those used in B8. We continue to use in-house lookup tables of absorption coefficients (ABSCO) parameterized as a function of temperature, pressure, wavelength, and water vapor mixing ratio for each of the main absorbing gases in the OCO-2 bands: $O_2$, $CO_2$, and water vapor ($H_2O$). Successive versions of these tables have been refined by incorporating new laboratory results and theoretical models for increasingly accurate absorption coefficients. The ABSCO version used in the B8 algorithm is ABSCO v5.0

(Drouin et al., 2017; Oyafuso et al., 2017); B7 used the previous ABSCO version, v4.2.

The ABSCO v5.0 $O_2A$ band tables represent a major step forward from previous ABSCO versions. Earlier ABSCO versions integrated the highest quality spectroscopic input from a range of studies that had focused on fitting different parameters independently. (See, for example, Thompson et al. (2012) and references therein). The ABSCO v5.0 tables are based on self-

15 consistent multispectral fits to laboratory spectra that include line mixing, speed-dependent Voigt line shape parameters, and collision-induced absorption (CIA). This self-consistency, and the use of laboratory spectra covering a range of pressures, temperatures and measurement techniques, are key features of the approach. The $O_2$ spectral line parameters, line mixing and CIA used in ABSCO v5.0 are described in Drouin et al. (2017). Parameters for broadening of $O_2$ by $H_2O$ are from the study by Drouin et al. (2014).

The impact of the latest multi-spectrum fitting update in the $O_2A$ band is shown in terms of the accuracy of the retrieved surface pressure in Figure 6. Panel (a) shows the retrieved surface pressure minus the prior for ABSCO version 4.2, which was used in version 7 of the algorithm, while panel (b) shows the same for ABSCO v5.0, used in version 8. The main improvements seen are that the retrieved surface pressures in version 8 are essentially unbiased with respect to the meteorological prior over

25 land, and that the land and ocean differences are reduced and centered closer to zero. We also note that for OCO-2, no additional line strength scaling was required in the $O_2A$ band, as has been necessary for all previous ACOS/GOSAT versions (see e.g. Crisp et al., 2012). For ACOS-GOSAT B3.5 retrievals, an $O_2$ scaling factor of 1.0125 was found to be beneficial, perhaps because of slight instrumental differences between OCO-2 and GOSAT.

The ABSCO v5.0 tables for the 1.61 and 2.06 $\mu$m $CO_2$ bands use line parameters and line mixing models derived from self-consistent, multispectral fits by Devi et al. (2016) and Benner et al. (2016), respectively. The parameters are derived from fits to laboratory spectra at multiple pressures and temperatures and the computation incorporates a speed-dependent Voigt line profile with nearest-neighbor line mixing. Earlier versions of the ABSCO tables (Benner et al., 1995; Devi et al., 2007) were based entirely on room temperature multi-spectrum fitting, with theoretical temperature dependences of the line shape and

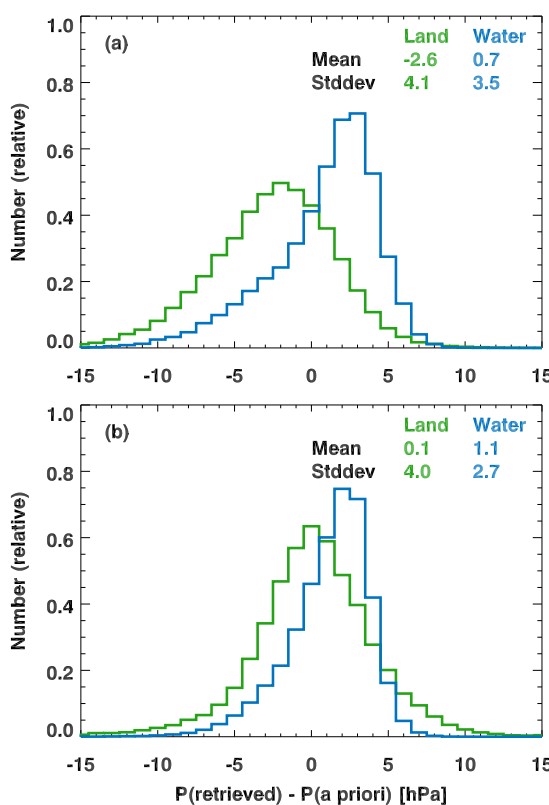

**Figure 6.** Retrieved minus prior surface pressure for a large selection of OCO-2 soundings, using both the oxygen-A band spectroscopy model from (a) ABSCO v4.2 and (b) ABSCO v5.0, as described in the text. ABSCO v5.0 spectroscopy leads to a more consistent retrieval of surface pressure over both land and ocean surfaces.

line mixing parameters. The updated spectroscopy includes analyses of spectra recorded at temperatures from 170K to 296K, representing a significant advance. Parameters for broadening of $CO_2$ by $H_2O$ are from Sung et al. (2009) for the 4.3 $\mu$m $CO_2$ band and extrapolated to OCO-2's $CO_2$ bands. Validation of the ABSCO v5.0 tables using up-looking TCCON spectra is described in Drouin et al. (2017) and Oyafuso et al. (2017). We note one important difference between the reference databases and our $CO_2$ absorption coefficients. We found it necessary to incorporate additional absorption in the center of the 2.06 $\mu$m band. This additional absorption was parameterized in order to reduce errors in retrievals with TCCON up-looking spectra. Further details can be found in Thompson et al. (2012) and Oyafuso et al. (2017).

Because the laboratory spectra underlying ABSCO currently are only good to roughly 1% absolute accuracy of line intensities, the algorithm allows for overall scaling factors for each of the two $CO_2$ bands. For the 1.61 $\mu$m band, the ABSCO v5.0 tables include a uniform scaling to bring the intensities from the Devi et al. (2016) multispectrum fit into line with reference

intensity measurements (estimated accuracy $\sim$0.2 %) from Polyansky et al. (2015). Oyafuso et al. (2017) show that this pre-scaling of the ABSCO using reference laboratory measurements results in good consistency between single-band up-looking $X_{CO_2}$ retrievals from ground-based FTS spectra and the $X_{CO_2}$ values reported by TCCON (which are themselves calibrated to agree with reference airborne profiles). Reference intensity measurements are not available for the 2.06 $\mu$m band at the current time. In tests within the OCO-2 Level 2 algorithm, using OCO-2 radiances, a scaling of 1.004 for the 2.06 $\mu$m ABSCO table was found to yield the best agreement between single-band retrievals performed using this band compared with single-band retrievals performed using the 1.61 $\mu$m ABSCO table as described above.

Finally, ABSCO v5.0 tables incorporate $H_2O$ line parameters from the HITRAN 2012 compilation (Rothman et al., 2013). We use an unofficial, modified version of the MT_CKD continuum, supplied by Eli Mlawer (Mlawer et al., 2012). This continuum version offers a compromise between previous versions of MT_CKD and measurements by Ptashnik et al. (2011), and falls relatively close to measurements by Mondelain et al. (2013). The subsequently-released MT-CKD 3.2 has been tested and shown to be a modest improvement over the unofficial version incorporated into ABSCO v5.0, with negligible changes to $X_{CO_2}$, but noteworthy improvements to the water column determination. Methane is not currently included in the B8 (or previous versions) of the forward model, as the impact of methane absorption was found to be negligible for $X_{CO_2}$ retrievals performed using the OCO-2 spectral ranges.

## 3.3 Residual Fitting using EOFs

ACOS B3.3 introduced a new way to deal with large spectral residuals caused by imperfect spectroscopy, solar model and instrument characterization, which were previously treated using a simple "empirical noise" parameterization (Crisp et al., 2012). In contrast, the new approach fits scaling factors to fixed spectral residual patterns for each band.

These patterns are the empirical orthogonal functions (EOFs) that result from a singular value decomposition of spectral residuals from training retrievals. Training scenes were selected to be largely devoid of cloud and aerosol effects, such that residual patterns due to unfitted clouds and aerosols are not a large contributor to the resulting EOF patterns. The EOFs are constructed such that the residuals $\mathbf{r}_{s,b}$ of each sounding $s$ and band $b$ can be approximately represented as a linear combination of the EOF patterns:

$$\mathbf{r}_{s,b} = \sum_{j=1}^{N_{eof}} c_{j,s,b} \, \mathbf{e}_{j,b} \tag{3}$$

where the vectors $\mathbf{e}_{j,b}$ are the EOFs for each band. For a diverse set of training retrievals, a matrix $\mathbf{M}$ is created for each spectral band and populated by the residuals of the spectral fits within that band. Training sets typically included more than 10,000 soundings.

Each matrix $\mathbf{M}$ is then decomposed into its eigenvectors using traditional singular value decomposition:

$$\mathbf{M} = \mathbf{UWV^T} \tag{4}$$

with the columns of $\mathbf{U}$ spanning an orthonormal basis of the most persistent spectral residual vectors observed in the training dataset. By convention, the first eigenvector explains the largest fraction of the total variance, as indicated by descending order of singular values (the diagonal elements of $\mathbf{W}$).

Application of this EOF technique substantially reduces the spectral residuals, yielding values of the relative RMS of the residuals of $\sim 0.1\%$ for each band, and reduced $\chi^2$ values near unity. For GOSAT, only the first EOF was found to be necessary. For OCO-2 B7 (and B8) target mode observations, better agreement with TCCON $X_{CO_2}$ was found when the largest three EOFs were employed. The first two EOFs for GOSAT, as well as the first three EOFs for OCO-2, are shown in Figure 7 for each spectral band. The EOFs for each of the eight spatial footprints sampled by OCO-2 are extremely similar, though they are solved for independently due to the slightly different spectroscopic response of each. The first EOFs for OCO-2 and GOSAT are very similar for each band, indicating that common forward model errors such as spectroscopy and top-of-atmosphere solar flux, rather than instrument-specific effects, are driving the EOF patterns. The first EOF is also very similar to the mean residual pattern, and typically accounts for 50-60% of the variance in the residuals. The second and third EOF typically account for only 1-3% of the variance in the residuals, with higher order EOFs accounting for even less. For the $O_2A$ band, the second EOF appears as a Doppler shift of the first. In the weak $CO_2$ band, the second EOF appears to be due to water vapor lines, while the third EOF appears to be a Doppler shift of the first. Higher order EOFs often exhibit additional instrument artifacts (such as unidentified bad spectral samples) and forward model errors related to water vapor, as well as other effects that are difficult to interpret.

The EOF formulation was modified in B8 in several ways. First, the EOFs were defined in terms of radiance per unit noise rather than pure radiance, in order to be consistent with the cost function metric that is minimized during the retrieval itself. However, the structure of the EOFs in B8 was much the same as in B7. Second, improved filtering of spectral samples that are contaminated by noisy or dead detector pixels greatly reduced their impact on the EOF patterns. Finally, a manual re-ordering of the EOFs was performed for each of OCO-2's eight spatial footprints, because the standard ranking by variance would occasionally flip the EOF patterns in different footprints. This mattered because sometimes the third and fourth EOFs would change places (and the current algorithm only fits the first three EOFs). This ensured that, to the extent that the EOFs of each footprint roughly matched, the same three EOF patterns for each footprint are fit within the L2FP retrieval.

## 3.4 Surface Model

The forward model for ACOS L2FP retrievals uses one of two surface models, depending on the location of the footprint. Water surfaces are simulated as a linear combination of a Cox-Munk ocean surface (Cox and Munk, 1954) and Lambertian reflector. This surface has seven parameters: wind speed, and a Lambertian albedo at a reference wavenumber with a linear

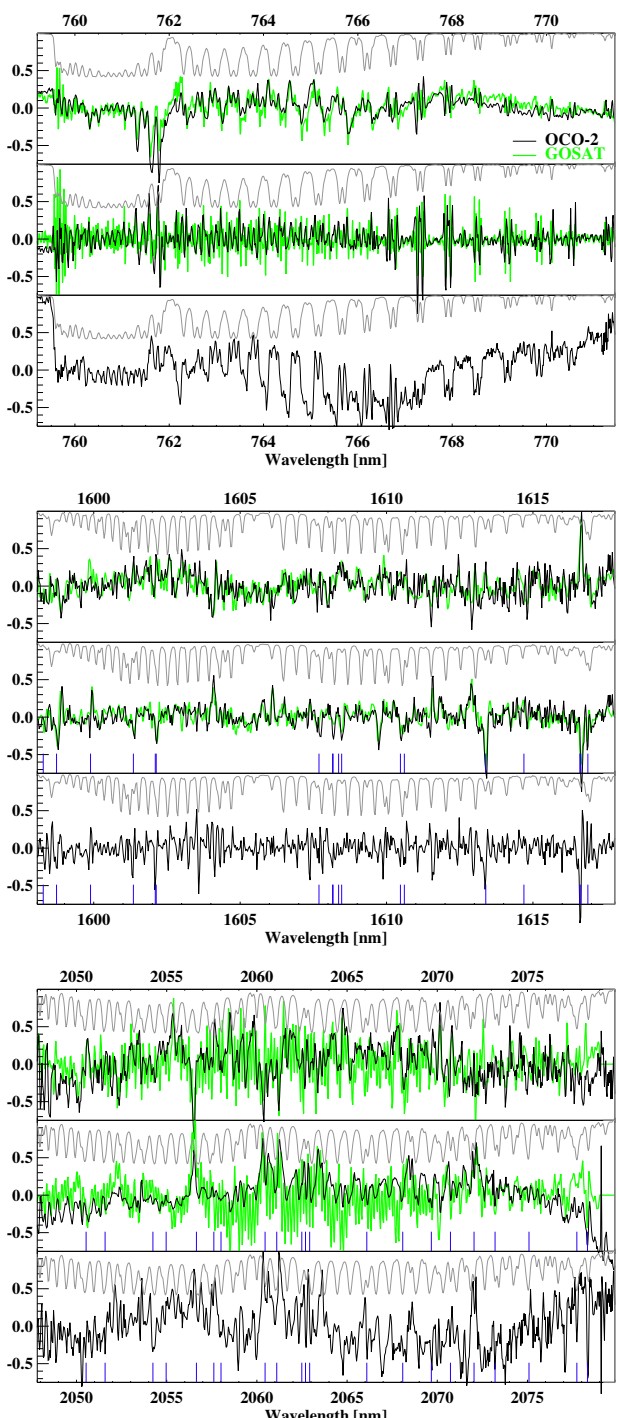

**Figure 7.** Spectral patterns of the EOFs for the $O_2$A band (top set of panels), weak $CO_2$ band (middle set of panels), and strong $CO_2$ band (lower set of panels) for GOSAT B3.5 (green) and OCO-2 B7 (footprint 4 only) (black). For reference, the light grey trace in each panel shows the modeled spectrum (not to scale). Some stronger water vapor absorption lines (blue vertical lines) in the weak and strong $CO_2$ bands correlate with features in the 2nd and 3rd EOFs.

spectral slope term in each of the three spectral bands. The prior wind speed is taken from the resampled meteorology (either ECMWF or GEOS5 FP-IT, as discussed in the next section), as for the other meteorological parameters. The strong $CO_2$ band Lambertian albedo is fixed to 0.02; the other six terms are fit in an essentially unconstrained fashion. This approach leads to the fitted Lambertian albedos generally staying small and positive, the latter of which is currently required by our radiative

transfer module.

Through ACOS B7, land surfaces were assumed to be purely Lambertian, with an albedo and albedo spectral slope retrieved for each band. The Lambertian surface assumes that the bidirectional reflectance distribution function (BRDF; the ratio of the radiance in the reflected direction to the irradiance from the incident direction; Schaepman-Strub et al. (2006)) is a constant

that is often specified as a scalar albedo. Since independent fits are done within each of the three OCO-2 spectral bands, this yields six state variables for land footprints.

Analysis of B7 OCO-2 target mode observations showed that the retrieved Lambertian albedo and aerosol optical depth sometimes exhibited dependence on the sensor zenith angle for observations of the same surface location. This indicated that

the true surface BRDF has dependence on the observation angles. A more physically justified approach would use a non-Lambertian model for the surface BRDF. For trace gas retrievals, a Lambertian surface assumption introduces no errors in the absence of multiple scattering between the surface and atmosphere; in this case, the retrieved albedo is interpreted as the surface reflectance at the primary scattering geometry (sun-surface-satellite). However, over brighter surfaces with some atmospheric scattering, the assumed BRDF could in principle affect the retrieval via the interaction of the retrieved aerosol,

surface pressure, and gas concentrations. Therefore, in B8 it was decided to change the surface model for land footprints to a non-Lambertian surface model. This model assumes a fixed BRDF shape and is assumes the surface is azimuthally symmetric, but allows for spectral dependence of the amplitude between and within each of our three bands; full details of the BRDF model are given in Appendix B. While this model does often show reduced correlation between view zenith and the retrieved BRDF amplitude, the retrieved $X_{CO_2}$ (as well as most other state vector parameters) shows very little change versus a version

of the B8 retrieval run with a Lambertian surface. Therefore, while B8 does use a non-Lambertian BRDF parameterization, a Lambertian surface appears to work equally well. This fact may be a consequence of the strong filtering used in B8, which tends to remove soundings with multiple scattering. Future applications of the ACOS L2FP algorithm to cases with higher AOD may be more strongly impacted by the non-Lambertian BRDF.

### 3.5  Additional Retrieval Algorithm Changes

In addition to these changes, a number of additional (mostly minor) changes have also been made to the ACOS L2FP algorithm since B2.9. In B2.10, the prior $CO_2$ profile was changed to match that used by TCCON, which was more realistic than our previous prior formulation; as of B8, this corresponds to the GGG2014 version (Toon and Wunch, 2014). Generally speaking the TCCON $CO_2$ prior profile is relatively simple: it is a function of latitude, altitude, and date only. It includes a simple formulation of the seasonal cycle and currently assumes a fixed secular increase of 0.52%/yr (or 2.08 ppm/yr at 400

**Table 5.** Median $\pm 1\sigma$ of GEOS5 FP-IT - ECMWF differences for GOSAT soundings passing the ABP cloud filter.

| Variable | Land | Ocean |
|---|---|---|
| Surface Pressure (hPa) | $0.07 \pm 0.73$ | $0.05 \pm 0.43$ |
| T2m (K) | $0.4 \pm 2.7$ | $0.3 \pm 0.6$ |
| T@700 hPa (K) | $0.0 \pm 0.8$ | $-0.2 \pm 0.8$ |
| TCWV (kg/m$^2$) | $0.1 \pm 1.8$ | $0.4 \pm 2.4$ |
| Surface Wind Speed (m/s) | $0.4 \pm 1.3$ | $-0.4 \pm 0.9$ |

ppm). There is no land/ocean or other meridional dependence. It requires specifying the tropopause height, and has simple formulations for the profile in the boundary layer, free troposphere, and stratosphere. A small mistake in the $X_{CO_2}$ averaging kernel was also fixed in B2.10; this was caused by inconsistent assumptions regarding the pressure-dependent gas absorption cross sections throughout our retrieval code, which led to an obvious "kink" in the averaging kernel that had long been visually evident (see e.g. Figure 2 of Connor et al., 2008). We use the 2016 version of the Toon solar transmittance spectrum[2] (Toon, 2014). Changes in the prior covariance matrix for $CO_2$ (O'Dell et al., 2012) were also considered, but rejected, as tests using alternate covariance matrices showed insignificant performance improvements.

In B3.3, solar-induced chlorophyll fluorescence (SIF) fitting over land surfaces was introduced. This change was introduced to combat a bias in $X_{CO_2}$ that results from not fitting for fluorescence when it is present, due to its impact on the $O_2A$ band. This problem and our fluorescence fitting scheme are described in detail in Frankenberg et al. (2012). Briefly, we fit for the mean and slope of the fluorescence at the top-of-canopy as a function of wavelength; the mean is expressed as a fraction of the continuum radiance level at a wavelength of 755 nm. The spectral dependence of the fluorescence is taken to be linear. Additional minor algorithm changes to B3.3 included reducing the prior surface pressure uncertainty from 4 to 1 hPa for GOSAT, which is likely to be a more accurate representation of the true prior surface pressure uncertainty for the majority of scenes (see e.g. Salstein et al., 2008). This has the added benefit of reducing the interference error between SIF, aerosols, zero-level offset and surface pressure.

In B3.4, fitting for SIF was turned off for GOSAT medium gain observations over land, as these regions are nearly all desert with very little biological activity. For OCO-2, the SIF prior is taken from the official SIF retrieval, as described in Sun et al. (2018), because SIF retrievals from individual soundings are meaningful for OCO-2 due to its relatively high SNR.

In B8, the prior height of the cirrus cloud layer relative to the surface pressure was moved slightly, from the fixed value of $x = 0.3$, to just below the tropopause height (which is a relatively strong function of latitude). The calculation of the tropopause

---

[2]Available at https://mark4sun.jpl.nasa.gov/toon/solar/solar_spectrum.html.

height itself was also refined in B8, which also improved the calculation of the prior $CO_2$ profile. Finally, the prior meteorology was changed in B8 from ECMWF to GEOS5-FP-IT (Suarez et al., 2008; Lucchesi, 2013). Some statistics regarding differences in surface pressure, temperature, water vapor, and surface wind speed between the two models for retrieved GOSAT soundings are given in Table 5; the corresponding difference statistics for OCO-2 soundings are nearly identical. Only soundings passing the $O_2A$ band prescreener are included. In general, the two models are very similar, with for instance 95% of all soundings having a surface pressure difference of less than 1.5 hPa. Surface pressure probably affects our retrieved $X_{CO_2}$ the most, as it is used not only in the retrieval, but also in the bias correction, where differences in the prior surface pressure will lead to a first order change in the bias-corrected $X_{CO_2}$. Currently then, the "noise" from the surface pressure difference between these two models would amount to roughly 0.6 hPa, or about 0.25 ppm in $X_{CO_2}$, which is quite a bit less than our noise-driven error ($\sim 1.4$ hPa on average) and regional biases ($\sim 2.4$ hPa on average). Retrieved surface pressure errors are discussed in more detail in Section 4.3.4.

## 4   Retrieval Filtering and Bias Correction

All soundings passing the prescreening criteria (Table 1) are processed with the L2FP retrieval algorithm. Of these, some 10-20% fail to converge to a solution, typically because of unscreened clouds or other factors that cannot properly be modeled in the retrieval. Some fail simply because of the nonlinear nature of the problem - in general, there is no perfect way to minimize the cost function. Of the 80-90% of soundings that do converge to a minimum in the cost function, typically 3-6 iterations are required.

Despite our best efforts to prefilter problematic soundings, there are inevitably some retrievals with $X_{CO_2}$ errors that exceed those predicted by theory. Ideally, the $X_{CO_2}$ errors would be normally distributed, with errors consistent with the $1\sigma$ a posteriori uncertainty on $X_{CO_2}$ from the retrieval (see e.g. Rodgers, 2000), but often there are retrievals with systematically biased $X_{CO_2}$ and/or larger-than-expected scatter. This problem is partially mitigated by applying a bias correction, which can reduce both scatter on smaller spatial scales and biases on larger spatial scales. However, problematic soundings still remain. A quality filtering procedure then attempts to remove these soundings with larger-than-expected differences from our truth metrics. For GOSAT, this process was described in O'Dell et al. (2012) and Crisp et al. (2012). The problem of biases is dealt with via a linear bias correction (Wunch et al., 2011a). In this section, we describe both filtering and bias correction procedures for $X_{CO_2}$ for B8 retrievals only, unless otherwise noted. A similar procedure was used for GOSAT data as well as OCO-2 B7, but the procedures were more mature and robust for B8.

### 4.1   Truth Proxy Training Data Sets

Both filtering and bias correction require a *training data set*, which consists of soundings for which we have both the OCO-2 retrieved $X_{CO_2}$ as well as a reliable, independent estimate of $X_{CO_2}$. The latter we call a "truth proxy". We used four such datasets: TCCON, models, models in the southern hemisphere only, and a new validation method for OCO-2 called the "small-

**Table 6.** $X_{CO_2}$ Truth proxies for retrieval evaluation.

| Name | $N_s$ Land | $N_s$ Ocean Glint | Details |
|---|---|---|---|
| TCCON | 228k | 88k | Geometric colocation requirement, GGG2014 |
| Multi-Model Median | 1132k | 861k | Median of 6 models (see text) |
| Models_SHA | 260k | 251k | same as above for lat<20S |
| Small Area Approximation (SAA) | 795k | 505k | areas < 100km along-track |

area approximation". Table 6 lists the truth proxies used in version 8, while Figure 8 shows the spatial distribution of the truth proxy datasets matched to actual OCO-2 soundings.

### 4.1.1 TCCON-based Truth Proxy

The most direct truth proxy is the comparison to TCCON, which currently has 25 operational stations globally, but with heavy
representation in North America, Europe, Asia, and Oceania. For the OCO-2 B8 evaluation, the latest version of TCCON retrievals was employed (GGG2014, Wunch et al., 2015). Many schemes have been used to match air masses observed by satellites to those viewed from TCCON stations. Examples include a geographic-centric scheme (Cogan et al., 2012; Inoue et al., 2013; Oshchepkov et al., 2013; Kulawik et al., 2016), a scheme based on the potential temperature at 700 hPa (Keppel-Aleks et al., 2011; Wunch et al., 2011b), model-based selection (Guerlet et al., 2013), and geostatistical selection (Nguyen
et al., 2014). These more sophisticated techniques were primarily used because GOSAT had fairly sparse data and required relatively loose matching criteria to yield sufficient numbers of matched observations. This is less of a problem with OCO-2 at lower and mid-latitudes, with its higher spatial sampling density. High latitude validation with TCCON remains challenging, where OCO-2 data are still sparse.

Table 7 lists the TCCON sites used as truth proxies in this work. Training data ranges correspond to the quality filtering and bias correction procedures described in Sections 4.2 and 4.3, respectively. Validation data ranges correspond to the basic validation described in Section 5. Our colocation requirements for B8 were similar to those used for Wunch et al. (2017), in which we required that OCO-2 footprints were within 2.5° latitude and 5.0° longitude of the TCCON station, and that the observations occurred within 2 hours of each other. These requirements were modified slightly for the Caltech, Armstrong, and
Tsukuba stations in order to discriminate satellite observations taken over the nearby megacities of Los Angeles and Tokyo. Because of additional station data and a longer training period, there were roughly twice as many station-months of valid colocations for B8 as compared to our B7 training (roughly 400 vs. 190 station-months).

We estimate TCCON colocation errors to be on the order of 0.5 ppm, due to both colocation errors and TCCON station-level
biases (Hedelius et al., 2017). Even with these small errors, TCCON is an incomplete validation source due to its limited spatial

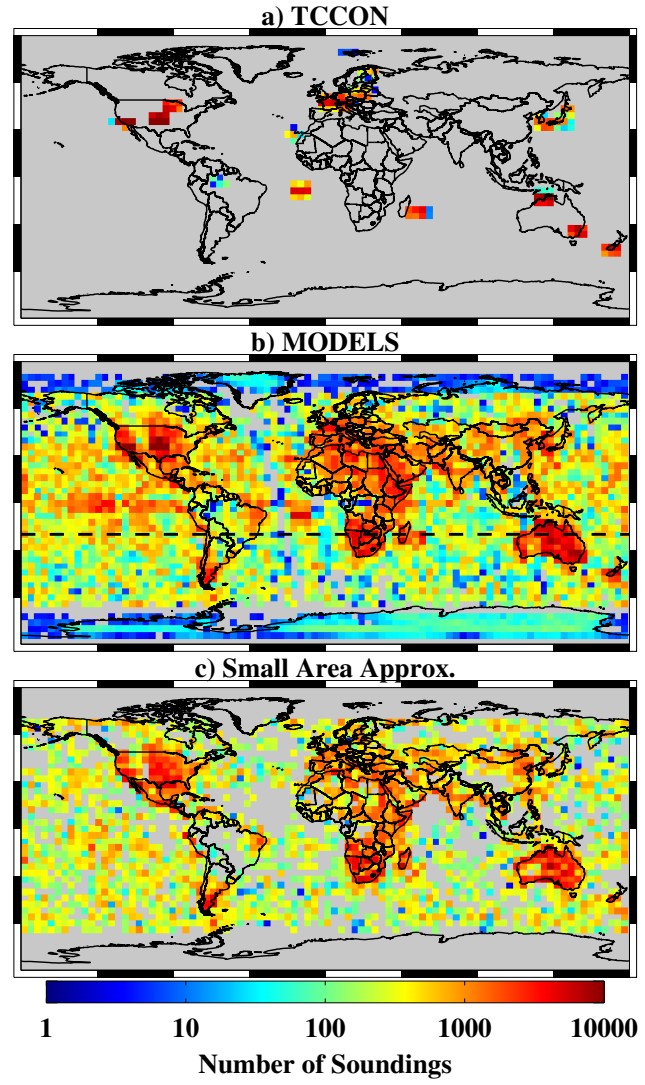

**Figure 8.** Sounding density of the truth proxy data in $4° \times 4°$ bins used in the OCO-2 version 8 $X_{CO_2}$ filtering and bias correction. The middle panel shows both the full global model-based truth proxy, and the southern hemisphere truth proxy as the portion below the dashed black line.

**Table 7.** TCCON stations used in this work.

| TCCON station | Training Date Range | Validation Date Range | Reference |
|---|---|---|---|
| Anmyeondo, South Korea | May 2015–Sep 2015 | May 2015–Aug 2016 | Goo et al. (2014) |
| Ascension Island | Sep 2014–Dec 2016 | Dec 2014–Feb 2017 | Feist et al. (2014) |
| Bialystok, Poland | Sep 2014–Jun 2016 | Sep 2014–Apr 2017 | Deutscher et al. (2015) |
| Burgos, Phillipines | Jan 2017 | Mar 2017–Apr 2017 | Velazco et al. (2017) |
| Bremen, Germany | Sep 2014–Jul 2016 | Sep 2014–Mar 2017 | Notholt et al. (2014) |
| Caltech, Pasadena, CA, USA | Sep 2014–Nov 2016 | Sep 2014–Feb 2017 | Wennberg et al. (2015) |
| Darwin, Australia | Sep 2014–Sep 2016 | Sep 2014–Oct 2016 | Griffith et al. (2014a) |
| Edwards (Armstrong), CA, USA | Sep 2014–Jun 2016 | Sep 2014–Aug 2016 | Iraci et al. (2016) |
| East Trout Lake, Canada | Jan 2017 | Oct 2016–May 2017 | Wunch et al. (2016) |
| Eureka, Canada | Jun 2015 | Aug 2015 | Strong et al. (2016) |
| Garmisch, Germany | Sep 2014–Aug 2016 | Sep 2014–May 2017 | Sussmann and Rettinger (2014) |
| Izaña, Tenerife, Spain | Dec 2015–Mar 2016 | Dec 2015–Jul 2016 | Blumenstock et al. (2014) |
| Karlsruhe, Germany | Sep 2014–Jun 2016 | Sep 2014–May 2017 | Hase et al. (2015) |
| Lamont, OK, USA | Sep 2014–Feb 2017 | Sep 2014–May 2017 | Wennberg et al. (2016) |
| Lauder, New Zealand | Sep 2014–Mar 2017 | Sep 2014–May 2017 | Sherlock et al. (2014) |
| Manaus, Brazil | April 2015–May 2015 | Nov 2014–Jun 2015 | Dubey et al. (2014) |
| Ny Ålesund, Spitzbergen, Norway | not used | May 2015–May 2017 | Notholt et al. (2017) |
| Orléans, France | Sep 2014–Oct 2016 | Sep 2014–May 2017 | Warneke et al. (2014) |
| Paris, France | Apr 2015–Mar 2016 | Oct 2014–Oct 2016 | Te et al. (2014) |
| Park Falls, WI, USA | Sep 2014–Dec 2016 | Sep 2014–May 2017 | Wennberg et al. (2014) |
| Réunion Island | Sep 2014–Nov 2016 | Sep 2014–May 2017 | De Mazière et al. (2014) |
| Rikubetsu, Japan | Oct 2014–Oct 2016 | Oct 2014–Feb 2017 | Morino et al. (2016b) |
| Saga, Japan | Sep 2014–Mar 2016 | Sep 2014–May 2017 | Kawakami et al. (2014) |
| Sodankylä, Finland | Oct 2014–Jul 2016 | May 2015–May 2017 | Kivi and Heikkinen (2016) |
| Tsukuba, Japan | Sep 2014–Feb 2017 | Sep 2014–May 2017 | Morino et al. (2016a) |
| Wollongong, Australia | Sep 2014–Nov 2016 | Sep 2014–May 2017 | Griffith et al. (2014b) |

coverage. For example, there are few stations in the tropics, none in the central Pacific or Central Asia, and, with the exception of Armstrong, in bright desert regions. Except when specifically stated, we employed the OCO-2 averaging kernel correction. A general treatment of averaging kernel corrections was first given in Wunch et al. (2011a). The specific correction we employ is taken from Nguyen et al. (2014), in which the TCCON retrieved profile is convolved with the OCO-2 column averaging
5 kernel before it is compared to OCO-2. This effect is generally smaller than 0.3 ppm in the column.

**Table 8.** Models used in this work.

| Name | Version | Land Biosphere | Inverse Method | Transport | Reference |
|------|---------|----------------|----------------|-----------|-----------|
| CAMS | 15r2 | ORCHIDEE | 4D-Var | LMDZ | Chevallier et al. (2010) |
| Univ. Edinburgh | v2.1 | CASA | EnKF | GEOS-Chem | Feng et al. (2009) |
| Jena CarboScope | s04_v3.8 | Special | 4D-Var | TM3 | Rödenbeck (2005) |
| CarbonTracker | CT2015, CT-NRT.v2016-1 | CASA | EnKF | TM5 | Peters et al. (2007), with updates documented at http://carbontracker.noaa.gov |
| TM5-4DVar | 2016 | CASA | 4D-Var | TM5 | Basu et al. (2013) |
| OU | 2016 | CASA | 4D-Var | TM5 | Crowell et al. (2018b) |

### 4.1.2 Small Area Approximation Truth Proxy

To supplement TCCON, we used a method new for OCO-2 called the "Small Area Approximation", or SAA[3]. The SAA relies on the high spatial resolution of OCO-2 footprints ($1.3 \times 2.3 \, \text{km}^2$), and the relatively long decorrelation length of $CO_2$ concentration in the atmosphere (500-1000 km, see e.g. Chevallier et al., 2017, Fig 1). Specifically, this approximation assumes

that for a given overpass of an area not larger than 100 km in spatial extent, $X_{CO_2}$ can be considered uniform over the area. True $X_{CO_2}$ variability was evaluated by Worden et al. (2017) by examining output from the GEOS-5 7x7 km$^2$ "nature run". It was found to be typically less than 0.1 ppm per 100 km areas away from strong known sources, thus justifying our small area assumption. In fact, this error is considerably lower than can be obtained by any of the other truth metrics. The major drawback of this method is that it is insensitive to biases due to variables that vary slowly on these small scales, such as those related to

viewing geometry and some surface and aerosol parameters.

### 4.1.3 Model-based Truth Proxies

The third validation dataset is based on results from global carbon flux inverse models, and is referred to as the "Multi-Model Median". In order to evaluate OCO-2 retrievals against a posteriori results from an array of models, and to avoid the biases in one particular model, a suite of 6 models sampled at the OCO-2 sounding locations and times was used. Table 8 provides a

summary of the models that were used. The models generally differed in their prior flux assumptions, prior flux uncertainty, transport model, initial conditions, spatial resolution, and inverse method, but had one commonality in that all assimilated in-situ $CO_2$ concentration data. Because of these differences, the models often yielded a posteriori $X_{CO_2}$ fields that disagreed to some extent, with differences ranging from a few tenths of a ppm to several ppm as discussed below. We used model output that covered a minimum period from September 2014 through December 2015, though a few models (CarbonTracker, TM5-

---

[3]Not to be confused with the South Atlantic Anomaly.

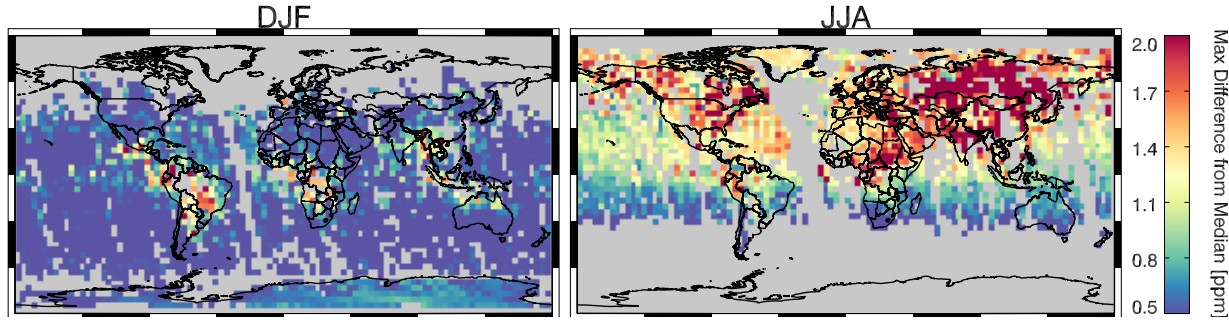

**Figure 9.** Maximum difference between each model and the model median in ppm, averaged over $4° \times 4°$ grid boxes. Two seasons are shown: DJF (left) and JJA (right). Soundings for which all models are within 1.5 ppm of the model median are retained in the model-based truth proxy.

4DVar) extended into March 2016. To compare against the models, for simplicity we computed only true $X_{CO_2}$ values from the a posteriori $CO_2$ concentrations, rather than averaging-kernel-corrected values. Previous authors have shown that this effect is typically small, on the order of a few tenths of a ppm (Wunch et al., 2011a; Inoue et al., 2013; Lindqvist et al., 2015).

5    For each matched OCO-2 sounding, the model median was computed from all available models for that sounding. If the any model $X_{CO_2}$ value differed by more than 1.5 ppm from the model median, that sounding was excluded from our training dataset. This requirement helped ensure that at the very least, all the models were generally consistent with each other for a given sounding in our training set. Generally the root-mean-squared difference of the model $X_{CO_2}$ values was less than $0.7$ ppm for any given sounding satisfying this requirement. The median of the model-predicted $X_{CO_2}$ for soundings satisfying 10    this criterion was then taken as the truth estimate.

Figure 9 shows maximum difference from the model median for both the northern hemisphere winter (December, January, February; DJF) and summer (June, July, August; JJA). Most soundings passed our "model-agreement" requirement over ocean at all times and over land in DJF, where the bulk of the land biosphere is quiet and hence $X_{CO_2}$ is more robustly modeled. 15    In JJA, however, a substantial fraction of land soundings fail this test, in particular over northern hemisphere regions such as Asia. Tests showed that our results were not strongly sensitive to the agreement threshold chosen.

Finally, Wunch et al. (2011a) used a truth proxy called the "Southern Hemisphere Approximation" (SHA) in which it was assumed that the southern hemisphere (25°S-55°S) could be taken to be meridionally uniform in $X_{CO_2}$ at any given time, with 20    a latitudinal gradient of -1 ppm from 25S to 55S, and the change in mean $X_{CO_2}$ over time could be prescribed with a linear secular trend (taken to be 1.9 ppm/yr). This served reasonably well for the GOSAT retrievals at that time, which exhibited rather large errors. However, the SHA has the primary shortcoming that meridional anomalies can sometimes exceed 0.5-1.0 ppm,

and are typically larger over land versus ocean. We find that substituting the model median instead of the zonally-corrected mean used in Wunch et al. (2011a) results in error variances of the approximation 3-4 times lower, when comparing against any particular model as truth. Therefore, in order to maintain a connection to the truth metric of Wunch et al. (2011a), in this work we adopt the modified SHA called "Model_SHA". This is simply the model median, discussed above, but only used in the southern hemisphere below a latitude $20°$S.

## 4.2 Quality Filtering

The construction of the operational OCO-2 filtering and bias correction for B7 is described in detail in Mandrake et al. (2015), with updates for B8 described in an online user's guide (Eldering et al., 2017). The training procedure for both filtering and bias correction for these two versions followed a similar approach. Below, we discuss the filtering and bias correction for version 8 only, and make notes where version 7 differed significantly. The filtering procedure yields two quantities. The first is a binary flag denoted the $X_{CO_2}$ *quality flag*, which requires that a series of parameter-based tests are all passed. The second is a graded set of "warn levels", which assigns each retrieval an integer value from 0 (most likely to yield accurate $X_{CO_2}$) to 5 (least likely to yield accurate $X_{CO_2}$). A genetic algorithm (Mandrake et al., 2013) finds combinations of variables that are best at predicting variance reduction in $X_{CO_2}$ over both small areas ($\lesssim$ 10x80 km$^2$) and in the Southern Hemisphere (south of 25S). In this document, we focus only on the quality flag filtering.

Filtering is accomplished by first identifying variables that cause the largest $\delta X_{CO_2}$, where $\delta X_{CO_2}$ is defined as the retrieved - true $X_{CO_2}$, the latter of which is evaluated for a given truth proxy. This was done sequentially, by identifying the single variable responsible for the largest fraction of the variance in $\delta X_{CO_2}$. We then created a simple threshold-based filter for this variable. After application of the filter, this process was repeated multiple times until it appeared that the majority of problematic data were removed. Because bias correction affected this procedure, a preliminary filter set was first created, after which a preliminary bias correction was developed. The preliminary bias correction was then applied, $\delta X_{CO_2}$ was updated accordingly, and the filters were re-derived using this bias-corrected $\delta X_{CO_2}$. Generally this had only a minor effect on the filters, and often served to increase the fraction of data passed through filtering.

Selection of thresholds for particular filters was somewhat subjective: generally bias was regarded as more problematic than scatter, but both were considered. Variables were typically selected as filters if they were correlated with bias greater than about 0.5 ppm, or significant scatter (greater than about 2 ppm). The filtering variables and thresholds were derived separately for land (combined nadir and glint) and ocean soundings. The final values of the filtering thresholds for the $X_{CO_2}$ quality flag are given in Appendix A. Filtering variables selected and their thresholds were the same or similar, regardless of the particular truth proxy used.

An example of this sequential filtering approach is shown in Figure 10, which shows the $X_{CO_2}$ error vs. filtering parameters for nadir and glint land soundings, using TCCON as the truth proxy. Overall, the results were found to be robust for all our truth

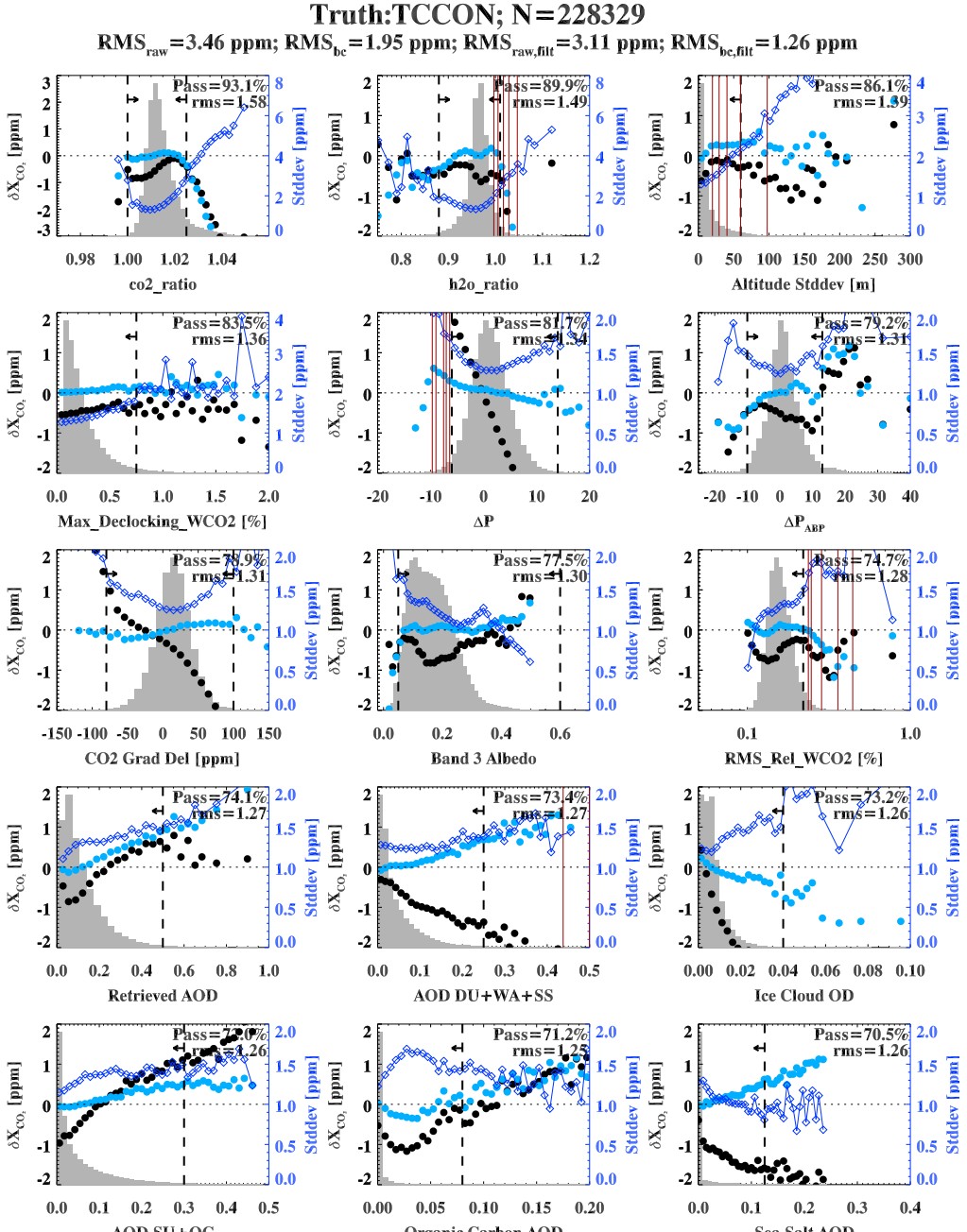

**Figure 10.** $\delta X_{CO_2}$ vs. select filtering variables for land (nadir+glint) data, using TCCON as a truth proxy. Shown are the mean bias in each parameter bin for both raw (black circles) and bias-corrected (light blue circles) $X_{CO_2}$, as well as the standard deviation of the bias-corrected $\delta X_{CO_2}$ (dark blue diamonds). The histogram of each parameter is shown in gray. The vertical black dashed lines denote filtering thresholds for the $X_{CO_2}$ quality flag, while the thin red solid lines show filtering thresholds for the warn levels. The quality flag filters are applied cumulatively from left to right and top to bottom. The fraction passing at each step, as well as the RMS error of the bias-corrected $X_{CO_2}$, are shown in the upper right corner of each panel. Please see Table A1 for a complete definition of all of the filter variables.

**Figure 11.** Same as Figure 10, but for ocean glint measurements, where the truth proxy is the multi-model median.

proxies. Just a few variables do the bulk of the filtering. For both land and ocean, the $CO_2$ and $H_2O$ ratios computed by the IMAP-DOAS preprocessor account for a significant fraction of the total filtering. These variables represent the ratio of the total column $CO_2$ ($H_2O$) as derived from the weak $CO_2$ band to that from the strong $CO_2$ band. As discussed at length in Taylor et al. (2016), values of these gas ratios that deviate significantly from unity indicate the presence of significant atmospheric

scattering. As shown in Figure 10, ratios significantly away from the median values can result in both large scatter and large biases. Another robust finding is that biases are associated with large absolute values of the retrieved - prior surface pressure (dP) for both the Level-2 and ABP preprocessor retrievals. All of these variables ($CO_2$ and $H_2O$ ratios and surface pressure) are most likely diagnosing scattering-induced errors due to improperly-modeled clouds and aerosols.

Two variables associated with small-scale variability are also associated with increased scatter: the standard deviation of the surface altitude within OCO-2's field-of-view, and another parameter called "Max_Declocking", which is determined independently for each of the three OCO-2 bands. The latter is related to a slope in the observed radiance within an individual sounding's field-of-view, and is determined from OCO-2's color slices as discussed in Crisp et al. (2017). The scatter associated with surface elevation appears to be related to an instrument-to-spacecraft offset specification error, which results in small

(several hundred meters) pointing errors, which is improved in the next data version (version 9) and allows for relaxation of this filter (Kiel et al., 2018).

Another interesting variable that can result in both bias and scatter is the tropospheric lapse rate of the retrieved $CO_2$ profile, called co2_grad_del. It is determined by the difference in retrieved $CO_2$ between the surface and the retrieval pressure level at

0.7 times the surface pressure, minus this same quantity for the prior:

$$\text{co2\_grad\_del} = [c(1) - c(0.7)] - [c_{ap}(1) - c_{ap}(0.7)] \tag{5}$$

where $c(x)$ and $c_{ap}(x)$ respectively denote the retrieved and a priori $CO_2$ dry air mole fraction at relative pressure $x$. The reason why this variable is strongly associated with bias and scatter is still being investigated; it may be due to $CO_2$ spectroscopy errors, or some other factor. There is also a filter associated with dark surfaces; scenes with a strong $CO_2$ band albedo

less than 0.05 consistently exhibit a bias in retrieved $X_{CO_2}$ and are thus excluded. Note this will tend to flag most snow- and ice-covered surfaces (such as over Greenland and Antarctica), which are highly absorbing at wavelengths longer than about 2 $\mu$m. It also tends to exclude dark forests such as in the Amazon. There are also filters associated with the retrieved slope of the strong $CO_2$ band albedo, the fit quality in the $CO_2$ bands, and a number of retrieved aerosol variables. Of particular note is the total retrieved optical depth associated with our larger aerosol types: dust, water cloud, and sea salt (DWS).

High values of DWS are associated with negative biases in $X_{CO_2}$ over land, and it is used as both a filter and bias-correction variable. Although ice is also a large type, it is confined to the upper atmosphere in our retrieval and has its own dedicated filter.

Similar variables are used for filtering over water surfaces (Figure 11), though note that almost no aerosol-related variables are used. This may be because water surfaces have relatively uniform optical properties, such that the retrieved variables in-

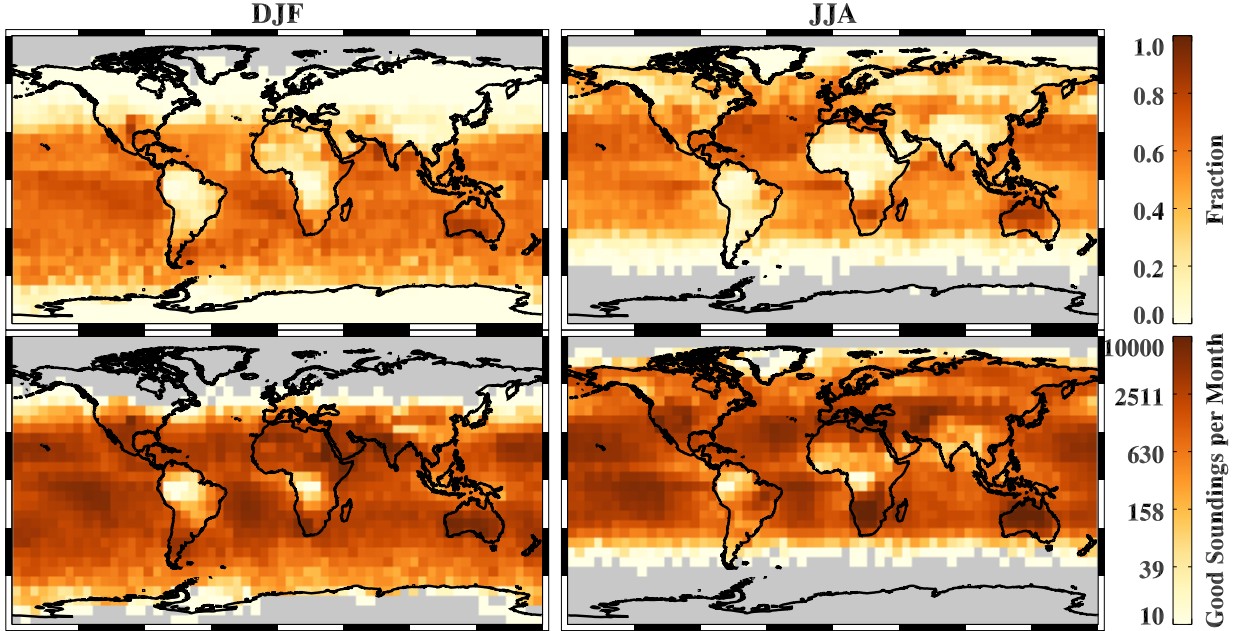

**Figure 12.** Fraction of L2FP processed soundings passing quality filter (upper row) and total number of good-quality soundings per month per $6° \times 6°$ boxes (lower row), for the northern hemisphere winter (DJF) and summer (JJA). The number plots have a logarithmic color scale, and grid boxes with no data are shown in grey.

directly associated with cloud and aerosol scattering, such as the $CO_2$ and $H_2O$ ratios and the slope of the strong $CO_2$ band albedo, are more effective than over land, obviating the need for additional aerosol-related filtering. It may also be because most downward-propagating, forward scattered light is absorbed by the ocean surface, so the pathways for aerosol contamination are significantly less than over land, as noted by Butz et al. (2013).

As seen in the upper left panel of Figure 11, the dominant filtering variable for water-glint soundings is the slope of the strong $CO_2$ band albedo. This is the slope of the retrieved Lambertian albedo in that band, which is generally small and is added onto the reflectivity coming from the primary Cox and Munk surface, which is a function of wind speed only. Negative slopes are strongly associated with $X_{CO_2}$ bias, which appears indicative of either cloud ice or sea salt aerosol scattering, both

10 of which yield a negative slope in these units[4]. Large positive values of this slope are likely associated with contamination by sulfate aerosol or other small particle types. The sensitivity of this variable to cloud and aerosol scattering has been confirmed with simulations. About 10% of water-glint soundings are flagged by this filter.

---

[4]The units of the albedo slope are in per unit wavenumber, increasing with wavenumber.

After filtration, about 31% of land soundings and 55% of water soundings pass the $X_{CO_2}$ quality flag [5]. As depicted in Figure 12, the pass rates are not uniformly distributed around the globe. Over land, very bright and dark surfaces are preferentially filtered out, as well as locations with many low clouds such as the Amazon, which are sometimes missed by our prefilters (Taylor et al., 2016). Nearly all soundings over ice surfaces are filtered out, because the albedo of ice is very low at $2\,\mu m$,

5 hence yielding low signal-to-noise. The higher quality of water soundings is likely due to higher uniformity of water surfaces in glint mode, higher and more uniform SNR in all three bands, and fewer surface-atmosphere scattering mechanisms. Over both land and water, soundings at higher solar zenith angles are also removed at a higher rate by our quality flag. This is most likely due to the relatively large effects of scattering on our retrievals for these geometries, specifically, when the fraction of the light received at the detector from atmospheric scattering is a larger fraction of the total. Over water, approximately 70%

10 of soundings pass at lower viewing angles, while nearly all soundings fail at high viewing angles.

**Table 9.** Land bias correction parameters, their coefficients and percentage of the variance explained, for different truth proxies and observing modes.

| Truth Proxy | Mode | N | Coefficent (%Variance) | | |
| --- | --- | --- | --- | --- | --- |
| | | | dP | co2_grad_del | DWS |
| TCCON | Nadir | 92k | -0.38 (33%) | -0.028 (17%) | -8.8 (4%) |
| | Glint | 68k | -0.38 (38%) | -0.026 (14%) | -6.4 (2%) |
| | Target | 245k | -0.29 (22%) | -0.023 (24%) | -7.8 (6%) |
| SAA | Nadir | 242k | -0.37 (38%) | -0.031 (26%) | -9.5 (10%) |
| | Glint | 251k | -0.36 (41%) | -0.030 (24%) | -9.6 (9%) |
| Models | Nadir | 281k | -0.34 (28%) | -0.029 (21%) | -9.3 (8%) |
| | Glint | 300k | -0.34 (30%) | -0.027 (19%) | -9.0 (8%) |
| Models_SHA | Nadir | 87k | -0.35 (28%) | -0.032 (29%) | -8.8 (8%) |
| | Glint | 87k | -0.36 (29%) | -0.029 (25%) | -10.3 (12%) |
| B8 Adopted | All | | -0.36 ± 0.028 | -0.029 ± 0.0027 | -8.5 ± 1.1 |
| Reference Value | | | 0.0 | 15.0 | 0.0 |
| B7 Adopted | All | | -0.30 | -0.028 | -7 to -11* |

*B7 used $ln$(DWS) rather than DWS in its bias correction.

---

[5]Note that these passing rates are lower than those in Figures 10 and 11, which were based on a smaller training dataset that included more successful and clear-sky soundings, with fewer soundings in difficult regions such as over heavy clouds or snow and ice-covered surfaces.

## 4.3 Bias Correction

After filtering, systematic biases remain in retrieved $X_{CO_2}$ which must be corrected in order to minimize errors. The OCO-2 bias correction contains three main pieces: parametric, footprint-level, and global biases. Parametric biases are functionally related to a given parameter associated with a given sounding. Examples of this could be surface pressure, albedo quantities, retrieved aerosol quantities, etc. Footprint-level biases are corrected to ensure that each of OCO-2's eight sensors, or "footprints", yield the same $X_{CO_2}$ value when observing similar scenes. This is not always the case due to small calibration errors in the eight individual footprints. The final step of the bias correction removes any global mean bias that may remain. The overall bias correction equation is then written as:

$$X_{CO_2,bc} = \frac{X_{CO_2,raw} - C_P(\text{mode}) - C_F(j)}{C_0(\text{mode})} \tag{6}$$

where $C_P$ is the mode-dependent parametric bias, $C_F$ is the footprint-dependent bias for footprints $j = 1\ldots8$, and $C_0$ represents a mode-dependent global scaling factor. The following subsections discuss each of these corrections in detail.

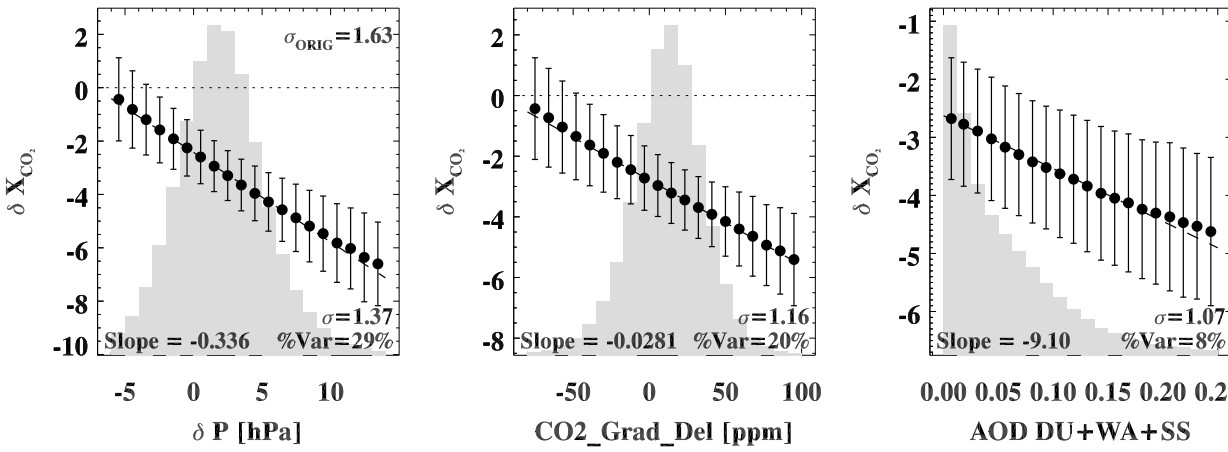

**Figure 13.** Multi-linear bias correction fit to the three variables used for land soundings. Here, land nadir and glint observations are shown, with the multi-model median truth proxy. The circles show mean values in each parameter (x-axis) bin, and the error bars are the 1-sigma standard deviations within each bin. The histogram shows the distribution of the parameter. The legend in each panel shows the starting and ending standard deviation after application of each variable, and the coefficient for that variable of the multiple regression using all three variables.

### 4.3.1 Bias Correction: Parametric Biases

The most complex but important of the three aspects of the bias correction is inferring biases dependent upon different retrieval parameters. Most near-infrared $X_{CO_2}$ retrievals have required this, for both GOSAT (Wunch et al., 2011a; Cogan et al., 2012;

Guerlet et al., 2013) and OCO-2 (Reuter et al., 2017; Wu et al., 2018) measurements. A nontrivial fraction of the bias comes from the retrieval algorithm itself, as shown in the simulation-based study of O'Dell et al. (2012), in which the instrument model and spectroscopy were perfect, yet biases still emerged in the retrievals. Previous versions of the ACOS algorithm applied to GOSAT have shown dependencies on the surface albedo in the $CO_2$ bands, $dP$ (retrieved minus prior surface pressure), co2_grad_del, the retrieved ice cloud height, and other variables. The parametric bias correction has the form of a multiple linear regression, following Wunch et al. (2011a):

$$C_P = \sum_i c_i (p_i - p_{i,ref}) \tag{7}$$

where $c_i$ are the regression coefficients, $p_i$ are the selected parameters, and $p_{i,ref}$ are convenient reference values. We note that the reference values are nontrivial in that they interact with the last term in the bias correction, the global scaling factor. Ideally, the reference value will be the value of the parameter at which that parameter does not bias the retrieved $X_{CO_2}$, but this is impossible to disentangle from the global scaling factor. Wunch et al. (2011a) took the parameter reference values to be the estimated global mean value of each parameter. Here we do not require this, though for some variables, the estimated global mean is used.

In order to identify the variables of interest, we used all four truth proxies and identified combinations of one, two, three, and four variables that removed the most variance, for each observing mode and over both land and water. Variables that remove less than 5% of the variance are not included, as overfitting is a potential danger here. Typically the different truth proxies agree on the most important variables, but disagree on the variables that explain just a few percent of the variance or less. As shown in Table 9, it was found that three fit parameters were required over land, and that their values did not strongly depend on observing mode. These variables were $dP$ (retrieved minus prior surface pressure), co2_grad_del, and finally DWS, which as stated previously is the combined retrieved optical depth of dust, water cloud, and sea salt aerosol. DWS represents the retrieved optical depth of large particles in the lower-to-middle troposphere in the retrieval. While ice cloud particles are large, they are placed in the upper troposphere in the retrieval, and all other aerosol types in the retrieval are much smaller.

In Table 9, the coefficients of each parameter inferred from each truth proxy and observing mode typically agree to within 10-20%. The final result represents a combination of the average of these individual values, but was also driven by consensus amongst the scientists involved. Table 9 also gives the approximate uncertainty on each parameter, which is estimated as the standard deviation of the estimates from the different truth proxies and viewing modes. Also shown is the B7 bias correction, which was very similar, though it used $\ln(DWS)$ instead of DWS. Figure 13 shows the result of the multiple regression for these three variables against the model-based truth proxy, for all nadir and glint soundings over land. In general, $dP$ explains about 30% of the variance over land, co2_grad_del about 20%, and DWS roughly 5-10%.

A similar procedure was followed for glint soundings over water. For this observing mode, only $dP$ and co2_grad_del were needed; all other variables explained only a very small fraction of the variance, and were not consistent among truth proxies.

**Table 10.** Ocean glint bias correction parameters for different truth proxies.

| Truth Proxy | N | Coefficent (%Variance) | |
| --- | --- | --- | --- |
| | | dP | max(co2_grad_del, -6) |
| TCCON | 72k | -0.24 (30%) | 0.078 (5%) |
| SAA | 385k | -0.22 (60%) | 0.092 (10%) |
| Models | 610k | -0.25 (35%) | 0.105 (8%) |
| Models_SHA | 157k | -0.12 (16%) | 0.134 (21%) |
| B8 Adopted | | $-0.23 \pm 0.06$ | $0.090 \pm 0.024$ |
| Reference Value | | 0.0 | -6.0 |
| B7 Adopted | | -0.08 | 0.077 |

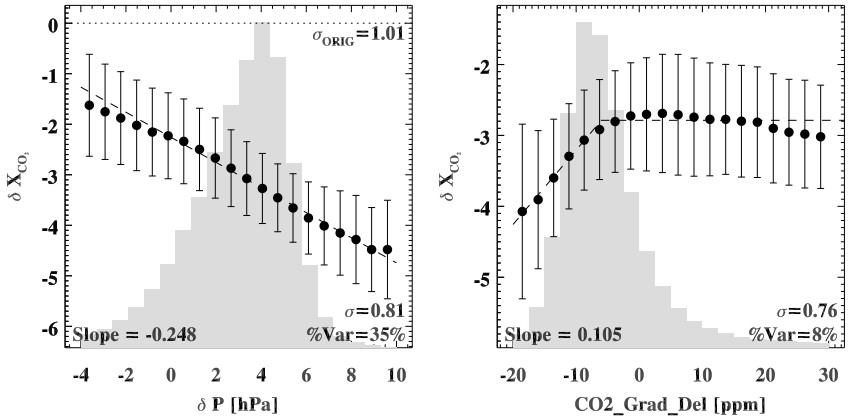

**Figure 14.** Same as Figure 13, but for ocean glint measurements again using the model mean truth proxy.

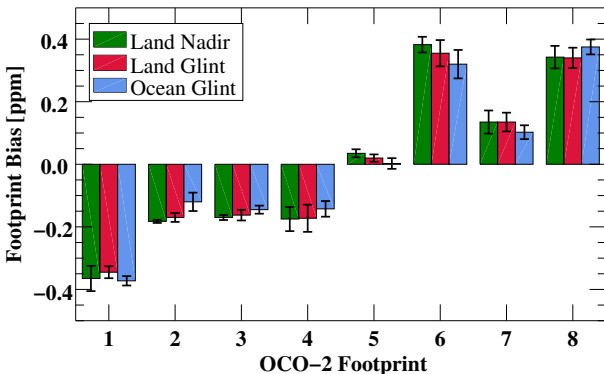

**Figure 15.** Estimates of the OCO-2 footprint biases, estimated separately for each observing mode and surface type. Because of their similarity, a single set of biases was used in the end.

The fit to these two variables for the model-based truth proxy is shown in Figure 14. A true linear regression will not work for co2_grad_del. Instead of fitting a nonlinear form, we instead fit against the variable max(co2_grad_del, -6). This gives essentially the fit as shown in the figure, where the best-fit line to the bias increases with increasing co2_grad_del until a value of -6, above which the fitted bias is held constant. As shown in Table 10, the different truth proxies again yielded similar results
to within about 20%, with the exception of "Models_SHA", which was an outlier. The reason for this is unknown, though we speculate that the actual parameter variability in the southern hemisphere is too small to obtain sensible slopes. Therefore, this truth proxy was excluded in the calculation of the final coefficients for glint soundings over water. Also, it is worth noting that the $dP$ coefficient in B7 was roughly three times smaller than the value of -0.23 adopted for B8. This was driven by inconsistencies in the B7 truth proxy data sets and a very small training data set, which yielded an unrealistically small value.
Later analyses showed that B7 probably should have used a higher value, more in line with the B8 result.

### 4.3.2  Bias Correction: Footprint Biases

After fitting for the parametric biases, the dependence of the $X_{CO_2}$ bias on footprint was evaluated. As with the parametric biases, the footprint biases were evaluated using the suite of truth proxies and for each observing mode separately (land nadir, land glint, and ocean glint). For all frames that contained all 8 footprints, the difference of each footprint from the mean of
its frame was calculated, with the result being the estimated set of footprint biases for each truth proxy. Note that this was done after application of the parametric bias correction. The resulting biases were quite consistent across truth proxies, thus the results across truth proxies were averaged. As shown in Figure 15, there was virtually no dependence on viewing mode, and no obvious land-water differences. It appears that the footprint-level biases are truly instrument related, and thus do not seem to strongly depend on other factors. Therefore, a single set of footprint-level biases was used. The adopted footprint biases for
footprints 1–8 were (-0.36, -0.15, -0.16, -0.14, 0.02, 0.33, 0.13, 0.34) ppm, with an uncertainty of roughly $\pm$ 0.03 ppm ($1\sigma$)

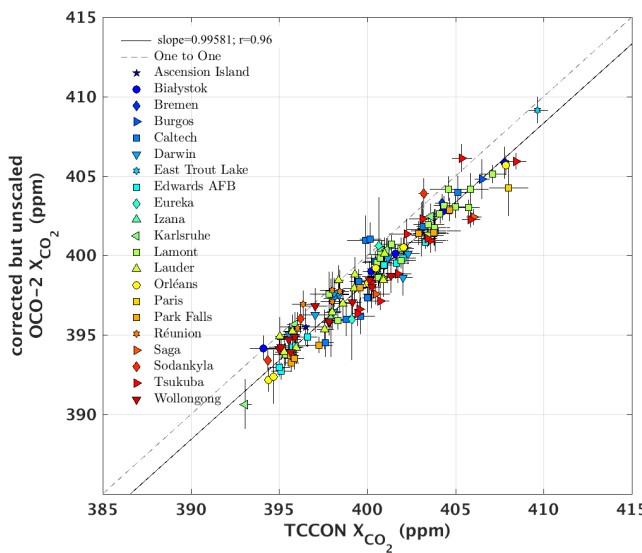

**Figure 16.** Scatter plot of 130 OCO-2 B8 target mode $X_{CO_2}$ observations versus colocated TCCON observations, used in the determination of the global scaling factor $C_0$ from equation (6). OCO-2 values have the parametric and footprint bias corrections applied.

on each. The reason for the general increase in this bias with increasing footprint is not known. Finally, while the fraction of the variance explained by footprint-level biases ($\sim$2%) is small compared to that explained by the parametric biases, they are straightforward to evaluate and could have an effect on local-scale analyses, and are therefore removed.

### 4.3.3 Bias Correction: Global Scaling

5   Despite the corrections described above, there is still an overall $X_{CO_2}$ bias on the order of 1-2 ppm relative to the true atmosphere. As shown in equation (6), the denominator term $C_0$ represents a global bias correction, and is parametrized as a function of viewing mode. As for the quality filtering and parametric bias correction, we found that the land scaling factor is roughly the same for nadir, glint, and target modes, such that a single scaling can be used for all land soundings. Ocean glint required a slightly different global scaling factor.

The B8 global scaling was determined primarily from several hundred direct OCO-2 overpasses of TCCON stations. We followed the geometric colocation method of Wunch et al. (2017), with the exception that sites in the southern hemisphere required the same latitude and longitude colocation thresholds as sites in the northern hemisphere. The TCCON value for a given overpass was determined as the mean of the observations within $\pm$2 hours of the OCO-2 overpass. At least 3 valid

15  TCCON and 20 quality-flag "good" OCO-2 soundings were required per overpass. For each viewing mode, the slope $m$ of the

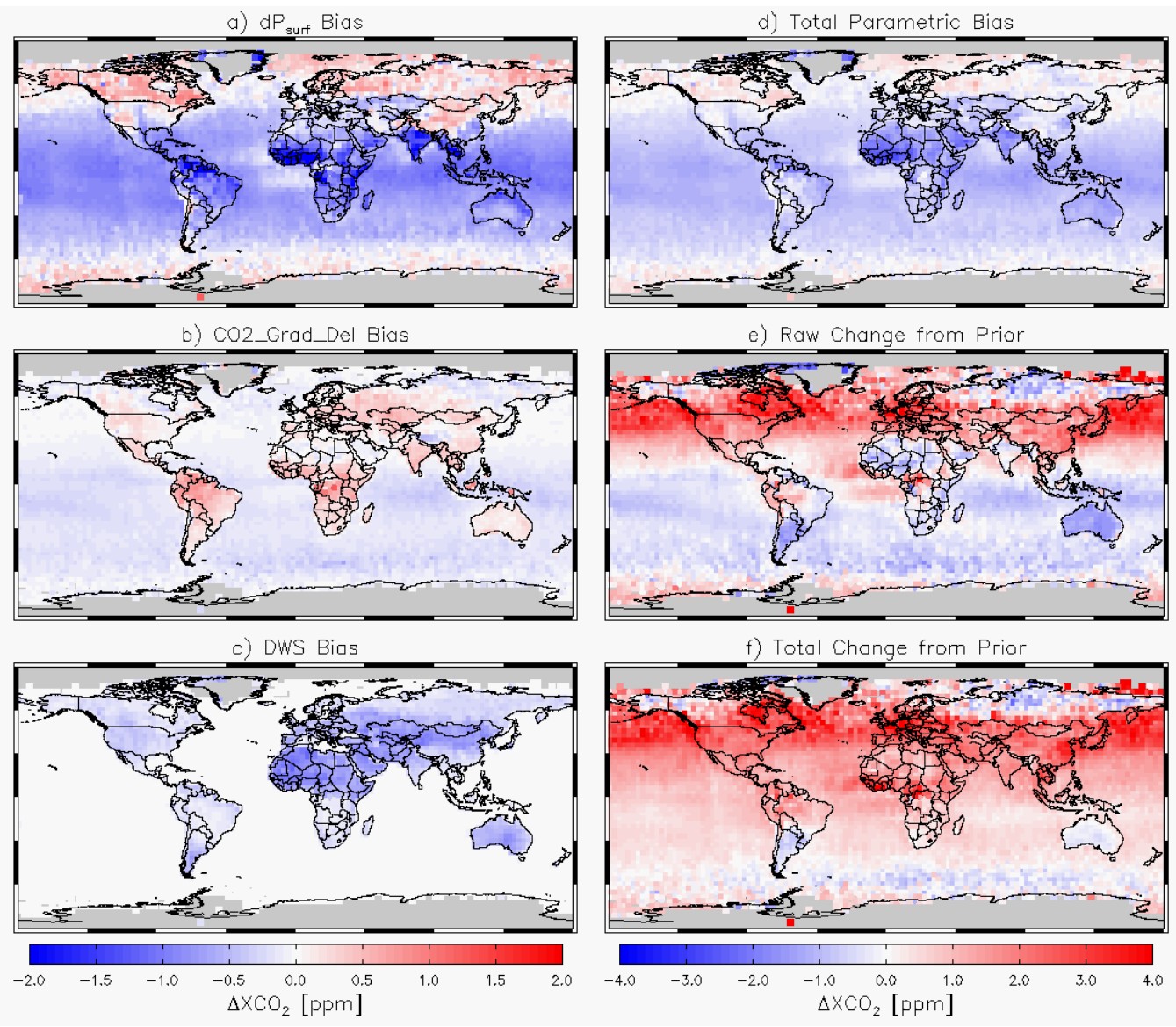

**Figure 17.** Annual mean biases and changes from the prior for all quality-flag "good" soundings from September 2014 through September 2017. (a)-(c) Parametric biases for $dP$, co2_grad_del, and DWS. (d) Sum of the parametric bias terms - note the color scale change. (e) Departure of the raw retrieved $X_{CO_2}$ including the global scaling correction, from the prior, $X_{CO_2,raw}/C_0 - X_{CO_2,ap}$. (f) Departure of the bias-corrected $X_{CO_2}$ from the prior, $X_{CO_2,bc} - X_{CO_2,ap}$.

**Table 11.** Global Scaling Factors (Divisors) for OCO-2 version 8.

| Data Set | Method | N | Divisor $C_0$ | |
|---|---|---|---|---|
| Land Target | TCCON | 138 | 0.9958 | |
| Land Nadir | TCCON | 313 | 0.9962 | |
| Land Glint | TCCON | 277 | 0.9960 | |
| Ocean Glint | TCCON | 236 | 0.9956 | *Using a value of 0.9958 for land soundings. |
| Ocean Glint | Coastlines* | 536 | 0.9955 | |
| Ocean Glint | Model Bootstrap* | 500,000 | 0.9954 | |
| Land (all) | Adopted | | 0.9958 | |
| Ocean Glint | Adopted | | 0.9955 | |

best-fit line passing through the origin was calculated with the method of York et al. (2004):

$$X_{CO_2,tccon} = \frac{X_{CO_2,OCO2}}{C_0} \tag{8}$$

which provides the best-fit value of $C_0$. The results for each viewing mode are shown in Table 11 under Method "TCCON". It can be seen that the three sets of land overpasses yielded a scaling factor $C_0$ consistent with each other to within their respective

errors. The errors are representative differences of the fitted slope due to both retrieval errors as well as linear fit differences (for instance, using a least-squares fit vs. a least-absolute-deviation fit), and were typically $\pm$ 0.0003 ($\sim$0.12 ppm). Because Target mode observations were better colocated, the Target observation value of 0.9958 was adopted for all land observations.

For ocean glint, the global scaling factor was estimated with three methods: direct overpasses of TCCON stations and two

independent "bootstrap" methods, using coastline-crossings and models. Direct overpasses of TCCON stations yielded a value of 0.9956, slightly lower than the adopted land value but still consistent to within errors. For the coastline bootstrap method, a set of several hundred small areas centered on coastlines were identified in which it was possible to ratio the mean value of $X_{CO_2}$ over water to the mean value of $X_{CO_2}$ over land. This yielded a mean water/land ratio of 0.9997$\pm$0.0001, meaning that ocean values were slightly lower than land values. Multiplying this water/land ratio by the land scaling value yielded an ocean

scaling value of 0.9955. Finally, the multi-model median truth proxy was used, wherein the slope of OCO-2 versus models was calculated, sampled at good quality OCO-2 sounding locations. It was found that land required a scaling of 0.9950 and ocean a scaling of 0.9946, suggesting a water/land ratio for OCO-2 of roughly 0.9996$\pm$0.0001, similar to that of the coastline crossing value, and thus an ocean scaling value of 0.9954. Note that the absolute comparison of models relative to OCO-2 was not used here, as spin-up issues and averaging kernel corrections (ignored in this analysis) could yield a spurious global

offset between the models and OCO-2. Therefore, only relative land-ocean differences were used to infer $C_0$ in this "Model Bootstrap" method. As seen in Table 11, the "bootstrap" methods were remarkably consistent with each other and with the

direct TCCON overpass method, all suggesting an ocean scaling of roughly 0.9955.

All of the methods described above for determining the global scaling of $X_{CO_2}$ show remarkable consistency with each other, giving confidence that the overall scale of OCO-2 data are known to within a few tenths of a ppm, and the land-ocean difference is known to better than 0.2 ppm. One caveat is that the direct TCCON overpass comparisons did not account for the averaging kernel correction in this analysis. After the release of OCO-2 $X_{CO_2}$ B8, subsequent analysis showed that this correction typically lowered the value of TCCON by $\sim$0.1 ppm on average relative to OCO-2. Therefore, there is additional possible uncertainty in the overall magnitude of OCO-2 data by this amount.

### 4.3.4 Bias Correction Evaluation

With all three sources of bias now characterized, the overall role of the bias correction in the retrieval can be evaluated. Figure 17 is an attempt to show this at the mean annual scale. Panels a-c show the biases due to the three different bias terms: $dP$, co2_grad_del, and DWS. Of these three, $dP$ is the strongest. This is apparent in panel (d), which shows the sum of the parametric biases (note the scale change between columns). Panel (e) shows the mean change from the prior in the raw retrieval, after correcting for the global bias term only (which is likely due to spectroscopic and instrumental bias). The change from the prior to the bias-corrected $X_{CO_2}$ is shown in panel (f), which combines these two terms.

The paramteric bias, while smaller than the difference between the raw retrieved and prior $X_{CO_2}$, is of a comparable order of magnitude. Ideally, the parametric bias would be much smaller. As noted above, the parametric bias is dominated by the $dP$ term. This term reflects the fact that the error in $X_{CO_2}$ is reduced using the prior rather than the retrieved surface pressure. That is, the ACOS-retrieved slant column of air is dominated by systematic errors that are not reflected in the estimated $CO_2$ slant column. We have identified two easily-correctable sources of such error, both related to how the B8 surface pressure prior fields were derived. One of these was caused by a slight error in the knowledge of the pointing of the OCO-2 instrument that induces spurious small-scale error in the estimated prior surface pressure in regions of high topographic variability. The second was caused by sampling the GEOS5 FP-IT surface pressure at the wrong time of day, by up to several hours. Both of these effects have been corrected for in the version 9 OCO-2 $X_{CO_2}$ data, described in detail in Kiel et al. (2018).

Even after correcting the prior, the OCO-2 surface pressure retrievals still contain significant regional biases. These biases have a largely zonal structure, with a negative bias of up to $\sim$ -5 hPa at the highest latitudes and a positive bias of several hPa in the tropics, with an overall mean bias of roughly +2 hPa. The standard deviation of the surface pressure bias (relative to the GEOS5 FP-IT prior) for quality-filtered soundings is roughly 2.8 hPa. By binning nearby soundings, it is found that greater than 2.4 hPa of this variation comes from systematic errors scales greater than $1°$. One possible hypothesis is that this systematic error is due to $O_2$ absorption cross section errors and how they manifest themselves in the retrieval. For instance, an incorrect parameterization of the temperature dependence of absorption could yield errors similar to those observed. Future

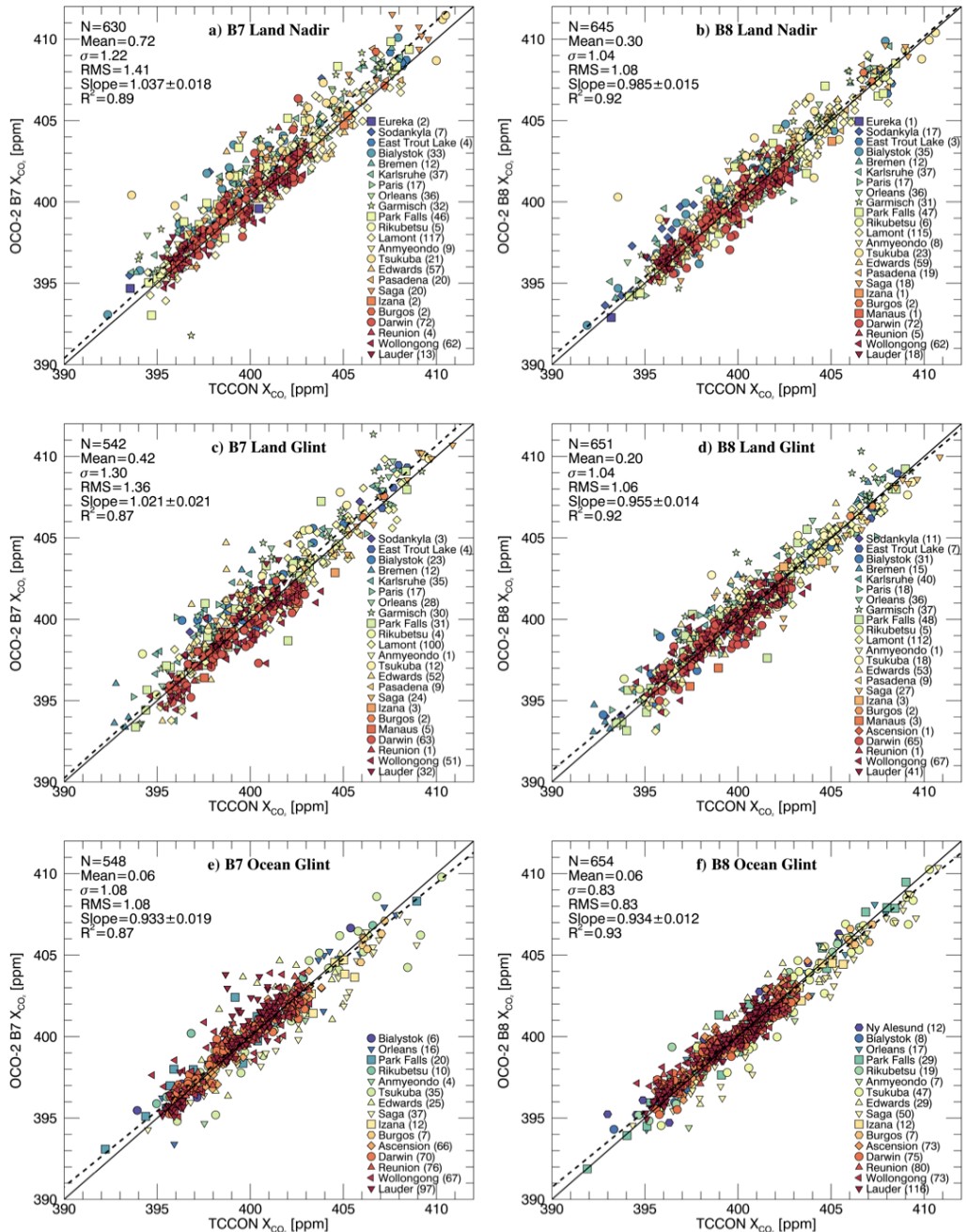

**Figure 18.** TCCON validation for OCO-2 $X_{CO_2}$ versions B7 (left column) and B8 (right column), for land nadir (top), land glint (middle), and ocean glint (bottom) observations. Each symbol represents the overpass-mean comparison for one site overpass, with the total number of overpasses per site given in parentheses. Thus each symbol represents tens to hundreds of OCO-2 observations co-averaged. Quality-filtered and bias-corrected $X_{CO_2}$ is shown for OCO-2, along with the averaging-kernel correction. The solid line denotes the one-to-one line, and the dashed line is the line of best fit. The colocation strategy is described in the text. Shown in the upper left of each panel are the total number of overpasses (N), the mean, standard deviation ($\sigma$), and RMS of the OCO-2 minus TCCON differences, the slope of best fit, and the $R^2$ of the two datasets.

updates to our spectroscopy may reduce this bias, and is an active area of research within the ACOS team.

Currently, however, these multi-hPa systematic errors in the retrieved surface pressure are likely larger than errors in the GEOS5 FP-IT prior, which are believed to be on the order of 1-2 hPa. For instance, three reanalysis surface pressure sets were recently compared and fund that the differences were as small as 0.5 hPa in the tropical oceans, and became larger at higher latitudes, such as in the southern ocean where RMS differences were on the order of 2 hPa (Jucks et al., 2015, section 4.2.1). This is the basic reason behind the artificially high (4 hPa) $1\sigma$ uncertainty on the prior surface pressure currently used in the OCO-2 retrieval. If future versions of the retrieval (including spectroscopy updates) yield a more unbiased retrieval of surface pressure, it may reduce the role of $dP$ in the bias correction and lead to a more accurate retrieval of $X_{CO_2}$.

The figure reveals several additional interesting features. The co2_grad_del bias is of a smaller magnitude than the $dP$ bias, and has the strongest effect in tropical forests, and a more diffuse effect elsewhere, including the tropical and southern oceans. The DWS bias has the largest effects over northern Africa, temperate Eurasia, and Australia, where large dust particles are prevalent. The total parametric bias pattern can sometimes cancel the departure from the prior, such as over Australia, indicating that the original departure was likely spurious. However, the overall large positive departure in the northern middle-to-high latitudes is not strongly affected by the bias correction, and thus appears to be a feature well-captured by the native retrieval itself. This latter feature seems to imply that the interhemispheric gradient is too weak in the TCCON prior. The GGG2014 TCCON $CO_2$ prior relies exclusively on the age of the air to create the interhemispheric gradient of $CO_2$. The next release of the TCCON software includes an age-independent term in the $CO_2$ priors representing the source/sink imbalance, which will roughly double the interhemispheric gradient (G. Toon, personal communication).

## 5  Brief Evaluation of OCO-2 $X_{CO_2}$

Thus far, we have completely described the mechanics of the current ACOS retrieval and methodology, but have yet to evaluate the actual algorithm performance. As B7 $X_{CO_2}$ was validated in detail in Wunch et al. (2017), in this section we focus primarily on the differences (mainly improvements) between B7 and B8. Figure 18 shows the relationship between colocated TCCON and OCO-2 $X_{CO_2}$ observations, for both B7 and B8 OCO-2 retrievals. Operational quality filtering and bias correction has been applied for each version. We used the same colocation strategy as described in Section 4.1.1, but with the additional requirement that fossil fuel emissions from the 1 km 2013 ODIAC database (Oda and Maksyutov, 2011), smoothed with a 5 km Gaussian smoother, be less than 300 g/m$^2$/month at the location of the OCO-2 soundings; this eliminated OCO-2 soundings in the vicinity of strong fossil fuel sources. All good-quality OCO-2 soundings within each overpass were averaged (thus, one symbol on each plot denotes one overpass). Only overpasses with at least 10 such OCO-2 soundings were included. The TCCON $X_{CO_2}$ values are averages of all good-quality TCCON soundings at that site within $\pm 2$ hours of the OCO-2 overpass.

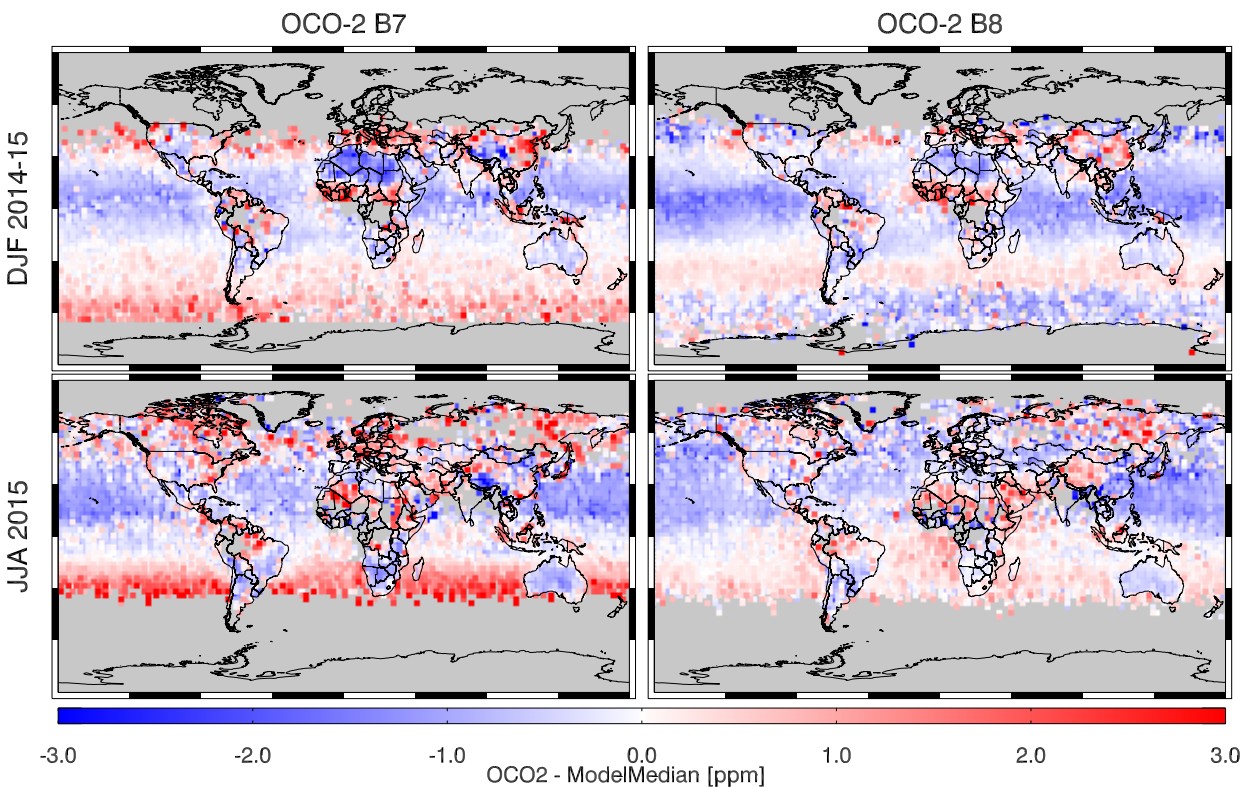

**Figure 19.** The difference between OCO-2 bias-corrected $X_{CO_2}$ vs. the model median where the models agree for both version 7 (left column) and version 8 (right column), using the agreement criteria as given in Section 4.1.3. Results are shown for two seasons: DJF 2014-15 (top row) and JJA 2015 (bottom row).

Based on these figures, B8 appears generally superior to B7 in terms of agreement with TCCON. First, there is more good-quality data in B7, at least for glint observations. This is primarily due to improvements in the prescreeners. In terms of accuracy, the scatter and outliers of $X_{CO_2}$ over both land and ocean are reduced in B7, especially over ocean for which the outliers were driven by southern hemisphere observations (in particular the high southern latitude sites, Lauder and Wollongong). The bias over land in B7 was significant, with a 0.3 ppm difference between land nadir and land glint. The overall apparent bias over land of 0.2-0.3 ppm is partly due to neglecting the averaging kernel effect when solving for the global divisors in the bias correction, and partly due to methodological differences in how we calculate the global bias in the first place. While the $R^2$ values are significantly higher than those reported in Wunch et al. (2017), this is due to the extended length of the data record used here. Coupled with the secular increase in $CO_2$, this leads to larger dynamic range in $X_{CO_2}$ and hence more signal and higher correlations than when using a shorter data record. Finally, we note that in B8 (and to some extent B7) there is generally a negative slope between TCCON and OCO-2. This appears to be due to a trend in OCO-2 $X_{CO_2}$ relative to TCCON (not shown), in which OCO-2 appears to be losing $X_{CO_2}$ at the rate of 0.1-0.2 ppm/year, though this trend is not statistically

significant. The apparent trend may be due to an OCO-2 calibration effect, and is currently under study.

The general improvement in B8 over B7 can also be seen in comparisons to models. Figure 19 shows the comparison of both versions to the 6-model suite as discussed in Section 4.1.3. The large-scale differences between OCO-2 and model $X_{CO_2}$ are generally reduced in the latest version. The positive bias with respect to models over the southern hemisphere mid-latitude oceans is nearly removed in B8, as are some negative biases over Australia and the Sahara desert. Positive biases in the northern hemisphere mid-to-high latitudes over land are also reduced. We also see that the latitudinal extent of the ocean glint data is greater in version 8, especially in DJF; again, this is primarily due to updates in the prescreeners. A large-scale negative difference between OCO-2 relative to the model median persists in the tropical oceans, particularly over the Pacific and Indian basins, and is currently under study.

There are clear deficiencies remaining in the B8 OCO-2 $X_{CO_2}$ product. First, note again that Figure 18 showed overpass average statistics. As each overpass had at least 10 OCO-2 soundings, and each sounding typically had $< 1$ ppm posterior uncertainty, if the errors were all independent, the overpass average errors would be less than 0.3 ppm. Even taking into account TCCON and colocation errors, it is likely that the overpass-level (=small area) errors are still much larger than this due to correlated errors in the OCO-2 retrievals. A more detailed validation of OCO-2 B8, along with how its errors are correlated and how they integrate down, is the subject of a forthcoming paper (Kulawik et al., 2018). Other error sources also are still plainly visible in the OCO-2 data. For instance, the topography-related biases noted by Wunch et al. (2017) still exist in B8, but have recently been tracked to a misspecification of the satellite-to-ground pointing vector, and will be corrected in a forthcoming version 9. Also, there are still cloud-related errors in the OCO-2 data, for instance as noted by Massie et al. (2016). It is believed that these are often related to 3-dimensional cloud effects, for instance as discussed in Merrelli et al. (2015).

## 6  Summary and Outlook

As described in this paper, the OCO-2 retrieval algorithm for $X_{CO_2}$ has been evolving more or less continuously over the last decade, and is beginning to achieve accuracies that enable ground-breaking carbon cycle science. The latest version (B8) has the lowest biases and highest throughput of any version yet, though regionally coherent biases still remain at a significant level ($\sim 1$ ppm). A number of choices and assumptions go into the algorithm, and due to its complexity in terms of the number of variables it must accommodate, further research is needed to improve it. These assumptions relate to a number of factors, such as clouds and aerosols, surface pressure, spectroscopy, potential instrument problems (e.g. scattered light, drifting calibration), to name a few. Advances in these areas will form the basis of future algorithm improvements. These issues affect not only OCO-2, but potentially all current and future sensors relying on this technology to measure $X_{CO_2}$ such as the TanSat (Yang et al., 2018), OCO-3 (Eldering et al., 2018), GOSAT-2 (Nakajima et al., 2012), MicroCarb (Pascal et al., 2017), and GeoCarb (Moore et al., 2018) missions.

To fully exploit space-based short-wave infrared measurements of reflected sunlight at high spectral resolution for studies of the carbon cycle, improvements must be made in several areas. First, satellite-based $X_{CO_2}$ retrieval biases must be further reduced. Next, in order to make and validate retrieval improvements, we must have validation data that are more accurate than the satellite retrievals in the first place. Satellite retrievals are currently pushing that limit; for instance, the stated station-level calibration accuracy for TCCON is currently 0.4 ppm (Wunch et al., 2010). Calculations of OCO-2 station-level differences with TCCON show the that mean absolute bias at all stations with at least 5 valid overpasses is 0.4 ppm, right at this level. Finally, we must have inversion and data assimilation systems that can make maximum use of these data. This requires, for example, minimizing transport model error, currently an active area of research (see e.g. Basu et al., 2018; Schuh et al., 2018).

Significant progress has been made in the past decade in the retrieval of $X_{CO_2}$ from SCIAMACHY, GOSAT, and OCO-2 radiances. This paper shows that this progress continues. Given the numerous sensors planned for development and launch in the near term, the future of passive remote sensing of $CO_2$ remains bright.

*Code and data availability.* The OCO-2 L2 Full Physics Code is open source and available on Github https://github.com/nasa/RtRetrievalFramework, and the User's Guide for it is available at http://nasa.github.io/RtRetrievalFrameworkDoc/. All of the OCO-2 data products are publicly available through the NASA Goddard Earth Science Data and Information Services Center (GES DISC) for distribution and archiving (http://disc.sci.gsfc.nasa.gov/OCO-2; OCO-2 Science Team, 2015). TCCON data were obtained from the TCCON data archive hosted by CaltechDATA, and are available from https://tccondata.org/.

## Appendix A: $X_{CO_2}$ quality flag definitions

The ACOS $X_{CO_2}$ quality flags use a number of variables. Each has an upper and lower threshold. These variables and thresholds are given in Table A1.

## Appendix B: Land Surface BRDF parameterization

Since OCO-2 observations outside of target mode contain only single observation geometries, it is unrealistic to attempt to retrieve the BRDF shape on a per-observation basis. This fact, combined with the similar improvement seen in the different trial BRDFs, suggested that a single fixed BRDF shape could be used for all land footprints. The selected BRDF shape is a particular parameter set for the Rahman-Pinty-Verstraete (RPV) kernel (Rahman et al., 1993), that has been used as an initial guess for spectral multi-angle polarimetric aerosol remote sensing (Dubovik et al., 2011). This fixed BRDF shape is used within the physical forward model, and a similar set of two state variables is applied independently to each band to allow for the BRDF amplitude to have a linear spectral variation across the band. Since the RPV kernel assumes the surface is azimuthally symmetric, the absolute values of the azimuth angles are unimportant and the kernel function can be expressed in

**Table A1.** $X_{CO_2}$ quality flag definition for version 8.

| variable | meaning | land filter | ocean filter |
|---|---|---|---|
| co2_ratio | Ratio of Band 2 to Band 3 $CO_2$ column from IDP* algorithm | [1,1.025] | [0.997,1.018] |
| h2o_ratio | Ratio of Band 2 to Band 3 $H_2O$ column from IDP* algorithm | [0.88, 1.01] | [0.88, 1.01] |
| dP | Retrieved minus prior surface pressure [hPa] | [-6,14] | [-4,10] |
| $dP_{ABP}$ | Retrieved minus prior surface pressure from ABP algorithm [hPa] | [-10,13] | [-50,10] |
| co2_grad_del | Retrieved vertical gradient in $CO_2$ [ppm] (see text for details) | [-80,100] | [-20,30] |
| albedo_slope_sco2 | Retrieved slope of the Lambertian component of the surface albedo [$cm^{-1}$] | [$-1.8 \cdot 10^{-4}$, $10^{-3}$] | [$5 \cdot 10^{-6}$,$7 \cdot 10^{-5}$] |
| rms_rel_wco2 | Relative RMS of Band 2 fit residuals [%] | $< 0.22$ | $< 0.30$ |
| Max_Declocking_wco2 | See text for details | $< 0.75$ | $< 0.2$ |
| Max_Declocking_sco2 | See text for details | | $< 0.3$ |
| eof33rel | Retrieved relative amplitude of 3rd EOF of Band 3 | | [-0.3, 0.25] |
| windspeed | Retrieved surface wind speed [m/s] | | [1.5,25] |
| Altitude Stddev | Standard deviation of the surface elevation in the FOV [m] | $< 60$ | |
| Band 3 Albedo | Retrieved Albedo SCO2 band | [0.05,0.6] | |
| $S_{31}$ | Continuum signal band 3 rel. to band 1 | [0.03, 0.4] | |
| $\tau_{IC}$ | Retrieved optical depth of ice cloud | $< 0.04$ | $< 0.035$ |
| $\tau$ | Retrieved total aerosol+cloud optical depth | $< 0.5$ | |
| $\tau_{DU} + \tau_{WA} + \tau_{SS}$ | Retrieved optical depth of three large types (dust, water cloud, and sea salt) | $< 0.25$ | |
| $\tau_{WA}$ | Retrieved optical depth of water cloud | [0.0005,0.1] | |
| $\tau_{SS}$ | Retrieved optical depth of sea salt | $< 0.125$ | |
| $H_{IC}$ | Retrieved relative pressure height of ice cloud | [-0.5, 0.45] | |
| $\tau_{SU} + \tau_{OC}$ | Retrieved sulfate + organic carbon optical depth | $< 0.3$ | |
| $\tau_{OC}$ | Retrieved organic carbon optical depth | $< 0.08$ | |
| $\tau_{ST}$ | Retrieved stratospheric aerosol optical depth | $< 0.02$ | |

*IMAP-DOAS Preprocessor

terms of the azimuth angle difference. Thus, the BRDF model used in the algorithm is a function of three angular variables, and can be expressed as:

$$\rho(\theta_i, \theta_r, \Delta\phi) = [w + s(\nu - \nu_0)]F(\theta_i, \theta_r, \Delta\phi; \mathbf{C}) \tag{B1}$$

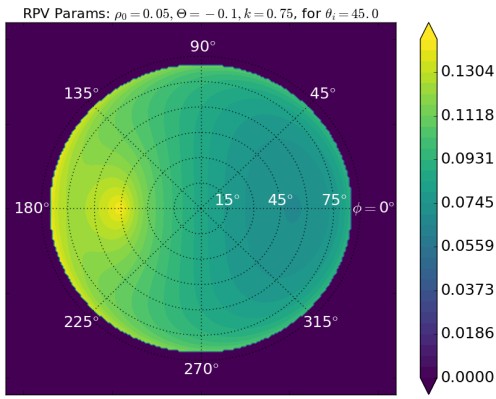

**Figure A1.** The shape of the RPV kernel for all reflection angles, for a $45°$ solar incidence angle. The incident direction is at $180°$. (meaning, the solar irradiance is directed inwards from the left side).

where $\theta_i$, $\theta_r$ are the zenith angles in the incident and reflected directions, $\Delta\phi$ is the relative zenith angle, $w$ and $s$ are the BRDF "weight" and "weight slope", $F$ is the fixed BRDF shape (the RPV kernel), and $\mathbf{C}$ are the fixed BRDF shape parameter values. The weight $w$ and weight slope $s$ are the retrieved variables, one each per band, with the linear variation computed in wavenumber space relative to per-band reference values ($\nu_0$). The RPV kernel function (Rahman et al., 1993) has three parameters, $\rho_0$, the hot spot parameter; $\Theta$, the asymmetry parameter; and $k$, the anisotropy parameter. The functional form, as implemented in the VLIDORT routines used by the OCO-2 forward model, uses the exact form as given in Rahman et al. (1993). The fixed values used for the parameters are: $\rho_0 = 0.05$, $\Theta = -0.1$, and $k = 0.75$. Figure A1 shows the RPV kernel shape for these parameters.

While the retrieval works with the weight and slope variables, $w$ and $s$, due to the normalization of the RPV kernel these are not the most convenient or intuitive quantities. Therefore, we evaluate the actual BRDF kernel function for the primary observation geometry (incident direction from the sun, reflected direction toward the sensor), and then scale by the retrieved weight and weight slope, to obtain the BRDF reflectance and reflectance slope. In the absence of atmospheric scattering, these values will be equal to the retrieved albedo and albedo slope using a Lambertian assumption. Therefore, in much of the high quality retrieval output from the Version 8 algorithm, the reported BRDF reflectance and reflectance slope values will be similar to the albedo and albedo slope values reported in the Version 7 results. Observations with relatively higher amounts of aerosols or other complicating effects would be expected to have larger differences between the BRDF reflectance and Lambertian albedo.

*Author contributions.* CO and AE were involved in nearly all aspects of this work. PW provided critical guidance on nearly all aspects of the work, throughout all stages. DC provided guidance on instrument effects on the algorithm and feedback on the algorithm. MG provided project leadership and algorithm guidance. BF designed and implemented many tests and performed substantial data analysis. CF was involved in earlier phases of algorithm development, in particular with the IMAP-DOAS retrieval, SIF retrieval, and EOF development.

MK contributed to the bias correction and analyzed TCCON target mode data. HL provided the updated cloud ice optical properties. LM created the testing datasets, warn levels, and contributed to filtering and bias correction. AM contributed substantial analysis and wrote the BRDF section. VN guided implementation of the BRDF surface model. RN performed data analysis, particular with regards to aerosol effects. GO led the target-mode operational effort and organized the validation activities. VP led the spectroscopy effort and wrote the spectroscopy section. TT edited early drafts of this manuscript and led the pre-filtering activities and analysis. DW provided critical analysis of

TCCON/OCO-2 comparisons and advice regarding TCCON data use. BD and FO made important contributions to the ABSCO spectroscopy. AC operationalized the code and assisted in daily operations. JM and MS coded the L2 algorithm and processed testing data sets. DB, SB, FC, SC, LF, and PP provided model data used in the filtering, bias correction, and validation efforts. MD, OG, DG, FH, LI, RK, IM, JN, HO, CP, CR, MKS, KS, RS, YT, OU, and VV provided TCCON data used in the filtering, bias correction, and validation efforts.

*Competing interests.* There are no competing interests present for the authors of this work.

*Acknowledgements.* Part of this work was conducted at the Jet Propulsion Laboratory, California Institute of Technology, under contract with the National Aeronautics and Space Administration (NASA) for the Orbiting Carbon Observatory-2 Project. Work at Colorado State University and the Geology and Planetary Sciences Department at the California Institute of Technology was supported by subcontracts from the OCO-2 Project. CarbonTracker CT2015 and CT-NRT.v2016-1 results provided by NOAA ESRL, Boulder, Colorado, USA from the website at http://carbontracker.noaa.gov. Jena CarboScope atmospheric inversion results provided by Christian Rödenbeck and the Max

Planck Institute for Biogeomchemistry, from http://www.bgc-jena.mpg.de/CarboScope/. Special thanks to Dr. Geoff Toon for illuminating discussions regarding the TCCON $CO_2$ prior. We also thank Dr. Julia Marshall and one anonymous reviewer for their reviews of this manuscript, which greatly improved its quality.

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
