# Peer review of "Improved Retrievals of Carbon Dioxide from the Orbiting Carbon Observatory-2 with the version 8 ACOS algorithm"

_Atmospheric Measurement Techniques, 2018_

## Referee Comment (RC1) · Anonymous Referee #1 · 21 Sep 2018

This paper, about the application of ACOS XCO2 retrieval algorithm to OCO-2 V8 L1b data, is a clearly written and well-conceived effort of value to the community. I suggest the publication of the paper. There are a few choices that were made in the study (described below) that I think need to be either addressed in more detail, or at least more fully justified. No extra data or processing are needed, so I think it will not be time-consuming. From a general aspect, I would suggest to short the paper a bit, for example the algorithm evolution review part distributed in several small sections and the bias correction part. Overall, this is a nice paper.

1 Page 2 Data prescreening: In this section, the fraction of soundings that passed the

prescreening is shown for December 2015 and June 2016. What is the overall fraction for each of these two months? Further, the variation of this fraction from month to month is also interesting since it can give us a general impression how many valid data we can have for leavel-2 processing.

2 Page 6 Table 2: The prior value used for $CO_2$ profile is not clear. It is only mentioned here that "same as TCCON". In page 19, line 27 it is also mentioned that "prior $CO_2$ profile . . . that used by TCCON". It is still not clear to me what is used as prior value. More explanations are needed since later the retrieval products are also validated with TCCON.

3 Page 19, line 20: Although is is mentioned in Appendix that the BRDF parameters is fixed, it is better to mention it here as well.

4 page 31 line 9: "31% of water soundings and 55% of land soundings pass the XCO2 quality flag", why water soundings have much lower chance to pass through the quality filtering? What is the overall data yields for ocean-glint and land respectively?

4 One major conclusion of the paper is that "Updates to the radiance calibration and retrieval forward model in version 8 have improved many aspects of the retrieved data products." This conclusion is made from the validation of bias-corrected V7 and V8 results with both TCCON (Figure. 18) and models (Figure. 19). Can we see similar imoprovement from uncorrected L2 data? If the improvements come from L1b radiance calibration and forward model itself, we should be able to see the effects from uncorrected L2 results. Also, I think looking at the uncorrected data will make things more clearly since the bias correction processing can confuse the source of improvements.

Some reference are lost, for example: Page 2, line 26: missed reference Page 5, line 10: missed reference Page 21, line 12: missed reference . . .

---

## Referee Comment (RC2) · J. Marshall (Referee) · 26 Sep 2018

This paper lays out the substantial work that has gone into the ACOS retrieval algorithm over the last several years, focussing on the adaptations in the version 7 and 8 builds tailored to OCO-2 data. It is quite an expansive paper as it contains information about several generations of the algorithm as applied to two different instruments, and marks the first peer-reviewed update of the full retrieval since 2012. The authors have done a good job of coherently structuring this information however, making it quite easy to follow. The manuscript is well-written and clear, and will provide a much-needed reference for the community of users already actively exploiting the resultant $XCO_2$ data.

[Figure]

The paper is optimally suited for publication in AMT, and is appropriate for publication once the authors have addressed a very few minor comments.

The main scientific questions that I was left with after reading the paper were related to the surface pressure differences. This difference between the retrieved and prior surface pressure leads to the dominant term in the empirical bias correction, and has quite a lot of structure, as is seen in Figure 6. The estimated uncertainty on the prior surface pressure was first decreased from 4 hPa to 1 hPa in B3.3, then increased to 2hPa in GOSAT B7.3, and then again to 4 hPa in OCO-2 B7. The discussion on P41 goes into some more detail on this, discussing some hypotheses about what the source of this difference could be, such as errors in the (temperature dependence of the) oxygen absorption cross section.

This plausible explanation, combined with the fact that the option of tightening the constraint on the prior is discussed, suggests that you are relatively certain that the 4 hPa uncertainty in the prior surface pressure is artificially exaggerated to allow for flexibility to account for this undefined bias source. This is implied in the discussion on P20, which suggests that a 1 hPa uncertainty is likely more realistic for the majority of scenes. This is again obliquely discussed at the end of Section 3, when discussing the differences between the surface pressure between the ECMWF and GEOS5 prior, which show only a difference of about 0.6 hPa, but this could also result from these two models being similarly biased and/or assimilating the same data.

It would be good to see the overinflation of the surface pressure prior uncertainty explicitly stated in the P41 discussion, along with a "best guess" estimate of what the true uncertainty in the prior surface pressure is. At present the information is all there, but the reader has to collect the information from several locations and piece it together. This point is likely relevant for other missions and retrievals as well, and can help users better interpret the data. Some further discussion into how you plan to tackle this identified problem, with or without improved $O_2$ spectroscopy, would be a welcome addition to Section 6.
**Typos/style comments:**

P2, L26: missing reference

P3, L3: as compared to: I would suggest changing this to"based on comparisons to" or even "when compared to".

P5, L10: missing reference

P5, L12: close parentheses after 3.1

P7, L3: a minor semantic point, but I would suggest "traits" rather than "behaviors"

P10, L11: You should define what ATBD stands for, or just use "as described in Crisp et al. (2010)."

Figure 3: The resolution could be better, will likely get picked up during editing.

P12, L8: evidence of similar → evidence of a similar

Figure 5 caption: The reference to the "operational retrieval" is confusing - which version is meant? I assume V7, but this should be clear.

P13, L32: Out → Our

There was some patchiness and inconsistency in the writing in the section describing ABSCO. Sometimes they were referred to as v5.0 or v4.0 (e.g. P14, L26; P15, L7), but usually just as 4.0, 4.2, or 5.0. On P15, L6, the second "the" should be removed. The paragraph starting at line 6 on P15 should perhaps be reworked, with the information incorporated in the two preceding paragraphs, which cover much of the same terrain (e.g. spectra at multiple temperatures in v5.0). Or perhaps I'm confused by this paragraph in general: "Because these spectra currently do not enable evaluation of intensities at accuracies greater than around 1

P17, L25: becomes → because

P20, L5: I would change "mean" to "the mean"

P20, L25: add comma after "correction"

P21, L13: missing citation

P21, L29: remove first "and"

P23, L1: Wasn't the potential temperature colocation scheme developed by Keppel-Aleks et al. (2011)? It doesn't appear in Wunch et al. (2011), or at least not in the Wunch paper you're citing there.

P23, L11: remove the first "were"

P23, table: missing reference for Sodankyla

P25, caption: 4x4° → 4°×4°

P26, L9: averaging-kernel corrected → averaging-kernel-corrected

P27, L6: data was → data were

P27, L31: I guess that the P in IDP already contains preprocessor. I had to go look it up again to be sure though, which raises the question: is it really worth definingh this TLA when it's only used twice in the main text? It could be defined separately in the caption of Table A1.

P27, L33: significant scattering present: I found this a bit awkward. Perhaps remove "present", or instead refer to "the presence of significant scattering in the atmosphere".

P28: In the caption it says the quality flags are applied cumulatively from top to bottom and left to right. Based on the fraction that passes each flag it appears left to right happens first, and then top to bottom. Please correct/clarify.

P30, L8-10: I'm a bit confused about how this pointing error in the instrument will be corrected in the next version. The next version of the instrument? Or the next version of the algorithm? If it's an instrument error, surely the algorithm will just be better taking it into account in the next version. Please clarify.

P30, L17: reason for this variable to be: Maybe better as "reason why this variable is"?

P30, L32: as has been noted (Butz et al., 2013). → as has been noted by Butz et al. (2013).

P33, caption: circles shown mean → circles show mean

P34, L14: is stated → as stated

P35, L16: line → in line

P38, Figure 17: There seems to be a large interhemispheric gradient associated with the correction to the prior, on the order of 3 ppm (in panel f). Could more information about the prior be given, besides the fact that it is the same as that used by TCCON?

P39, L15: "ocean was slightly lower than land": Please restate with all the necessary (semi-implicit) information, e.g. "the ocean values were slightly lower..." or similar.

P39, L15: yieled → yielded

P40, caption: "Overpass-mean level validation": I found this a bit confusing. What is "level" here?

P43, L9: over-pass → overpass

P44, L1: data that is → data that are

---

## Author Comment (AC1) · 12 Nov 2018

Dear AMT editor,

We'd like to thank Julia Marshall and the anonymous reviewer for their helpful reviews of our manuscript. We have responded to all of their critiques and made minor changes to the manuscript as a result. We believe the manuscript is significantly improved in quality.

Both reviewers noted several missing references in the typeset AMTD paper. This appears to have been a problem on AMTD's end, as all the references were included in the original paper version we sent to AMT. We'd like to work with AMT to ensure that typesetting of references is error-free in the final published version; please let us know how we can be of help in this process.

**Response to Reviewer 1 Comments**

*1 Page 2 Data prescreening: In this section, the fraction of soundings that passed the prescreening is shown for December 2015 and June 2016. What is the overall fraction for each of these two months? Further, the variation of this fraction from month to month is also interesting since it can give us a general impression how many valid data we can have for level-2 processing.*

We added the following sentences in section 2:

"In total, roughly 26\% of land soundings pass our pre-screener (28\% land nadir, 25\% land glint) and 27\% of ocean glint soundings pass it as well. Generally these fractions are strong functions of both location and time of year."

*2 Page 6 Table 2: The prior value used for CO2 profile is not clear. It is only mentioned here that "same as TCCON". In page 19, line 27 it is also mentioned that "prior CO2 profile . . . that used by TCCON". It is still not clear to me what is used as prior value. More explanations are needed since later the retrieval products are also validated with TCCON.*

We have updated the information on the TCCON prior, and now use this language in subsection 3.5:

"In B2.10, the prior $CO_2$ profile was changed to match that used by TCCON, which was more realistic than our previous prior formulation; as of B8, this corresponds to the GGG2014 version (Toon and Wunch, 2014). Generally speaking the TCCON $CO_2$ prior

profile is relatively simple: it is a function of latitude, altitude, and date only. It includes a simple formulation of the seasonal cycle and currently assumes a fixed secular increase of 0.52%/yr (or 2.08 ppm/yr at 400 ppm). There is no land/ocean or other meridional dependence. It requires specifying the tropopause height, and has simple formulations for the profile in the boundary layer, free troposphere, and stratosphere."

*3 Page 19, line 20: Although it is mentioned in Appendix that the BRDF parameters is fixed, it is better to mention it here as well.*

Done. Relevant text now reads: "Therefore, in B8 it was decided to change the surface model for land footprints to a non-Lambertian surface model. This model assumes a fixed BRDF shape and is assumes the surface is azimuthally symmetric, but allows for spectral dependence of the amplitude between and within each of our three bands; full details of the BRDF model are given in Appendix B."

*4 page 31 line 9: "31% of water soundings and 55% of land soundings pass the XCO2 quality flag", why water soundings have much lower chance to pass through the quality filtering? What is the overall data yields for ocean-glint and land respectively?*

Thankfully the reviewer caught this typo. The sentence should have read: "31% of land soundings and 55% of water soundings pass the XCO2 quality flag." We added an additional sentence to explain this difference: "The higher quality of water soundings is likely due to higher uniformity of water surfaces in glint mode, higher and more uniform SNR in all three bands, and fewer surface-atmosphere scattering mechanisms."

*4 One major conclusion of the paper is that "Updates to the radiance calibration and retrieval forward model in version 8 have improved many aspects of the retrieved data products." This conclusion is made from the validation of bias-corrected V7 and V8 results with both TCCON (Figure. 18) and models (Figure. 19). Can we see similar imoprovement from uncorrected L2 data? If the improvements come from L1b radiance calibration and forward model itself, we should be able to see the effects from uncorrected L2 results. Also, I think looking at the uncorrected data will make things more clear since the bias correction processing can confuse the source of improvements.*

The improvements are also evident in the $X_{CO2}$ results before bias correction. For instance in the TCCON comparison (e.g. Figure 18), the standard deviation of OCO-2 minus TCCON $X_{CO2}$, before bias correction, improves somewhat from 1.43 to 1.39 ppm; over ocean the improvement is 1.15 to 1.12 ppm. The improvement is larger after bias correction because of the worsened surface pressure retrievals in version 8, the effects of

which are largely removed via the bias correction. In the model comparison (e.g. Figure 19), the scatter before bias correction is improved somewhat in northern hemisphere winter between versions 7 and 8 (SD: 0.79 ppm → 0.76 ppm, for the grid box averages shown in the figure), but dramatically in northern hemisphere summer (SD: 1.07 ppm → 0.77 ppm). Thus in version 8, the scatter vs. models is now equivalent in summer and winter seasons even in the $X_{CO2}$ before bias correction, whereas this was not the case in version 7.

*Some reference are lost, for example: Page 2, line 26: missed reference Page 5, line 10: missed reference Page 21, line 12: missed reference . . .*

**Referee 2 comments**

*The main scientific questions that I was left with after reading the paper were related to the surface pressure differences. This difference between the retrieved and prior surface pressure leads to the dominant term in the empirical bias correction, and has quite a lot of structure, as is seen in Figure 6. The estimated uncertainty on the prior surface pressure was first decreased from 4 hPa to 1 hPa in B3.3, then increased to 2hPa in GOSAT B7.3, and then again to 4 hPa in OCO-2 B7. The discussion on P41 goes into some more detail on this, discussing some hypotheses about what the source of this difference could be, such as errors in the (temperature dependence of the) oxygen absorption cross section.*

*This plausible explanation, combined with the fact that the option of tightening the constraint on the prior is discussed, suggests that you are relatively certain that the 4 hPa uncertainty in the prior surface pressure is artificially exaggerated to allow for flexibility to account for this undefined bias source. This is implied in the discussion on P20, which suggests that a 1 hPa uncertainty is likely more realistic for the majority of scenes. This is again obliquely discussed at the end of Section 3, when discussing the differences between the surface pressure between the ECMWF and GEOS5 prior, which show only a difference of about 0.6 hPa, but this could also result from these two models being similarly biased and/or assimilating the same data.*

*It would be good to see the overinflation of the surface pressure prior uncertainty ex-plicitly stated in the P41 discussion, along with a "best guess" estimate of what the true uncertainty in the prior surface pressure is. At present the information is all there, but the reader has to collect the information from several locations and piece it together. This*

*point is likely relevant for other missions and retrievals as well, and can help users better interpret the data. Some further discussion into how you plan to tackle this iden- tified problem, with or without improved $O_2$ spectroscopy, would be a welcome addition to Section 6.*

The reviewer is absolutely correct on all of these points. Currently we overinflate the prior surface pressure uncertainty by a factor of ~4; our "best-guess" of the GEOS5 FP-IT surface pressure uncertainty is around 1 hPa (1sigma), based on comparisons to ground-based data and other models. We inflate it because of forward model error in our retrieval, which is primarily from spectroscopy, but possibly other error sources as well (for instance, imperfect characterization of the instrument ILS in the O2A band would also cause systematic errors in our retrieved surface pressure). The discussion of the retrieval surface pressure biases and their effect on XCO2 has been greatly expanded in the paper, and is now included in section 4.3.4.

*P2, L26: missing reference*

AMT needs to fix in typesetting.

*P3, L3: as compared to: I would suggest changing this to"based on comparisons to" or even "when compared to".*

Fixed.

*P5, L10: missing reference*

AMT needs to fix in typesetting.

*P5, L12: close parentheses after 3.1*

Fixed.

*P7, L3: a minor semantic point, but I would suggest "traits" rather than "behaviors"*

Changed as suggested.

*P10, L11: You should define what ATBD stands for, or just use "as described in Crisp et al. (2010)."*

Good point. Changed to the latter recommendation.

*Figure 3: The resolution could be better, will likely get picked up during editing.*

Fixed (on author's end), changed from png to eps. Also, it was noted in revisions by one of the co-authors (Merrelli) that the optical properties for Sulfate (SO) were incorrect in the original figure. This has been fixed.

*P12, L8: evidence of similar → evidence of a similar*
Fixed.

*Figure 5 caption: The reference to the "operational retrieval" is confusing - which version is meant? I assume V7, but this should be clear.*

Fixed.

*P13, L32: Out → Our*

Fixed.

*There was some patchiness and inconsistency in the writing in the section describing ABSCO. Sometimes they were referred to as v5.0 or v4.0 (e.g. P14, L26; P15, L7), but usually just as 4.0, 4.2, or 5.0. On P15, L6, the second "the" should be removed. The paragraph starting at line 6 on P15 should perhaps be reworked, with the infor- mation incorporated in the two preceding paragraphs, which cover much of the same terrain (e.g. spectra at multiple temperatures in v5.0). Or perhaps I'm confused by this paragraph in general: "Because these spectra currently do not enable evaluation of intensities at accuracies greater than around 1%".*

The troublesome paragraphs (written by several authors, leading to the patchiness noticed by the reviewer) have been reworked and cleaned up. All ABSCO version numbers are now referred to as ABSCO vX.x or ABSCO version X.x. The paragraph starting at line 6 on P15 has been shortened to remove the information covered in the preceding paragraph. It now is wholly about scaling the CO2 bands to yield consistent XCO2 retrievals from TCCON.

*P17, L25: becomes → because*

Fixed.

*P20, L5: I would change "mean" to "the mean"*

Fixed.

*P20, L25: add comma after "correction"*

Fixed.

*P21, L13: missing citation*

*P21, L29: remove first "and"*

Fixed.

*P23, L1: Wasn't the potential temperature colocation scheme developed by Keppel- Aleks et al. (2011)? It doesn't appear in Wunch et al. (2011), or at least not in the Wunch paper you're citing there.*

Changed reference to Keppel-Aleks (2011), "Sources of variations in total column carbon dioxide" in ACP.

*P23, L11: remove the first "were"*

Fixed.

*P23, table: missing reference for Sodankyla*

*P25, caption: 4x4° → 4°×4°*

Fixed.

*P26, L9: averaging-kernel corrected → averaging-kernel-corrected*

Fixed.

*P27, L6: data was → data were*

Fixed.

*P27, L31: I guess that the P in IDP already contains preprocessor. I had to go look it up again to be sure though, which raises the question: is it really worth definingh this TLA when it's only used twice in the main text? It could be defined separately in the caption of Table A1.*

Done.

*P27, L33: significant scattering present: I found this a bit awkward. Perhaps remove "present", or instead refer to "the presence of significant scattering in the atmosphere".*

Changed to "indicate the presence of significant atmospheric scattering".

*P28: In the caption it says the quality flags are applied cumulatively from top to bottom and left to right. Based on the fraction that passes each flag it appears left to right happens first, and then top to bottom. Please correct/clarify.*

Reviewer is correct, fixed.

*P30, L8-10: I'm a bit confused about how this pointing error in the instrument will be corrected in the next version. The next version of the instrument? Or the next version of the algorithm? If it's an instrument error, surely the algorithm will just be better taking it into account in the next version. Please clarify.*

Changed to "next data version (version 9)". Technically this is a change that happens at the L1B level, and thus flows down into preprocessors, meteorological resamples, and level-2 retrievals.

*P30, L17: reason for this variable to be: Maybe better as "reason why this variable is"?*

Adopted.

*P30, L32: as has been noted (Butz et al., 2013). → as has been noted by Butz et al. (2013).*

Changed.

*P33, caption: circles shown mean → circles show mean*

Fixed.

*P34, L14: is stated → as stated*

Fixed.

*P35, L16: line → in line*

Fixed.

*P38, Figure 17: There seems to be a large interhemispheric gradient associated with the correction to the prior, on the order of 3 ppm (in panel f). Could more information about the prior be given, besides the fact that it is the same as that used by TCCON?*

We have updated the information on the TCCON prior, and now use this language in subsection 3.5: "In B2.10, the prior $CO_2$ profile was changed to match that used by TCCON, which was more realistic than our previous prior formulation; as of B8, this corresponds to the GGG2014 version (Toon and Wunch, 2014). Generally speaking the TCCON $CO_2$ prior profile is relatively simple: it is a function of latitude, altitude, and date only. It includes a simple formulation of the seasonal cycle and currently assumes a fixed secular increase of 0.52%/yr (or 2.08 ppm/yr at 400 ppm). There is no land/ocean or other meridional dependence. It requires specifying the tropopause height, and has simple formulations for the profile in the boundary layer, free troposphere, and stratosphere."

Further, we analyzed the interhemispheric gradient and in conversation with Geoff Toon (primary author of the TCCON prior), he revealed that the interhemispheric gradient for CO2 is too small in GGG2014; this is being rectified in the next version. We added some text to the paper in the section discussing Figure 17 to this effect.

*P39, L15: "ocean was slightly lower than land": Please restate with all the necessary (semi-implicit) information, e.g. "the ocean values were slightly lower..." or similar.*

Changed to "meaning that ocean values were slightly lower than land values."

*P39, L15: yieled → yielded*

Fixed.

*P40, caption: "Overpass-mean level validation": I found this a bit confusing. What is "level" here?*

I agree this was a bit confusing, so I reworked the caption as follows:

"TCCON validation for \mbox{OCO-2} \xco\, versions B7 (left column) and B8 (right column), for land nadir (top), land glint (middle), and ocean glint (bottom) observations. Each symbol represents the overpass-mean comparison for one site overpass, with the total number of overpasses per site given in parentheses. Thus each symbol represents tens to hundreds of OCO-2 observations co-averaged. …"

*P43, L9: over-pass → overpass*

Fixed.

*P44, L1: data that is → data that are*

Fixed.